# Satformer: Accurate and Robust Traffic Data Estimation for Satellite Networks

**Liang Qin**
Xidian University
liangqin@stu.xidian.edu.cn

**Xiyuan Liu**[*]
Xidian University
markliu225@stu.xidian.edu.cn

**Wenting Wei**
Xidian University
wtwei@xidian.edu.cn

**Chengbin Liang**
Xidian University
chengbin@stu.xidian.edu.cn

**Huaxi Gu**[†]
Xidian University
hxgu@xidian.edu.cn

## Abstract

The operations and maintenance of satellite networks heavily depend on traffic measurements. Due to the large-scale and highly dynamic nature of satellite networks, global measurement encounters significant challenges in terms of complexity and overhead. Estimating global network traffic data from partial traffic measurements is a promising solution. However, the majority of current estimation methods concentrate on low-rank linear decomposition, which is unable to accurately estimate. The reason lies in its inability to capture the intricate nonlinear spatio-temporal relationship found in large-scale, highly dynamic traffic data. This paper proposes Satformer, an accurate and robust method for estimating traffic data in satellite networks. In Satformer, we innovatively incorporate an adaptive sparse spatio-temporal attention mechanism. In the mechanism, more attention is paid to specific local regions of the input tensor to improve the model's sensitivity on details and patterns. This method enhances its capability to capture nonlinear spatio-temporal relationships. Experiments on small, medium, and large-scale satellite networks datasets demonstrate that Satformer outperforms mathematical and neural baseline methods notably. It provides substantial improvements in reducing errors and maintaining robustness, especially for larger networks. The approach shows promise for deployment in actual systems.

## 1 Introduction

As a potential complement to terrestrial networks, satellite networks are envisioned to provide broadband connectivity with seamless coverage and in a cost-effective manner. Internet service and content providers are interested in satellite networks due to their wide international coverage and lower entry costs in rural and underdeveloped areas [1].

Traffic engineering [2, 3, 4] and topology engineering [5] of satellite networks, such as access control, routing and congestion control, are key to achieve efficient control of satellite networks, which rely on real-time perception of global traffic data [6]. Timely and accurate traffic measurements beyond basic metrics are undoubtedly beneficial for such applications.

However, it is troublesome and costly to collect massive traffic data by measuring all transmission pairs directly [7], since traffic data is naturally distributed throughout the entire network. In order to support the network operation of the emerging mega-constellations, there is an urgent need to

---

[*]First Author and Second Author contribute equally to this work
[†]Corresponding author

38th Conference on Neural Information Processing Systems (NeurIPS 2024).

explore cost-effective traffic measurement methods. Traffic data estimation is a feasible approach for large-scale satellite networks, where the global traffic data can be estimated according to partial traffic sampling and measurement [8].

Due to inherently dynamic natures of spatial distances and orbital positions in satellite networks, traffic volumes and patterns vary over time [9]. Emerging mega-constellations networks typically involve numerous satellites, so the dynamic traffic data between satellite pairs can be represented as high-dimensional matrices or tensors. This complexity makes it difficult to capture the complicated relationships within the data [10]. Furthermore, the instability of inter-satellite and satellite-ground links often leads to the loss of traffic data during transmission. It should also be noted that not all satellite pairs have constant communication demands. As a result, the measured traffic data is often sparse and incomplete, making traffic data estimation complex [1]. Therefore, the primary challenge in accurately and robustly estimating satellite network traffic data lies in effectively capturing the complex and nonlinear spatio-temporal correlations while maintaining robustness for varying sequence lengths [11].

Indeed, most efforts in traffic data estimation focus solely on low-rank linear decomposition, which cannot effectively capture the nonlinear spatio-temporal correlations among large-scale and dynamic traffic data, leading to inaccurate estimations. Therefore, developing a novel approach is crucial for enhancing traffic estimation performance to effectively extract and utilize the complex and nonlinear spatio-temporal correlations among inter-satellite traffic data.

For large-scale and highly dynamic satellite network traffic data, we propose Satformer, a new neural network architecture designed for accurate and robust traffic estimation. Satformer systematically constructs encoder-decoder components with stacked spatio-temporal modules to effectively capture complex spatio-temporal correlations in traffic data. Within each module, an adaptive sparse spatio-temporal attention mechanism (ASSIT) is adopted to extract key features from numerous sparse inputs by focusing on specific local regions. This enables Satformer to capture nuanced traffic patterns essential for accurate estimation. This is particularly useful in satellite networks where traffic may be concentrated in certain areas due to regional demand or satellite coverage. Additionally, ASSIT is more robust to sparsity as it can identify and focus on areas with higher data density, which may contain more informative traffic features, rather than being overwhelmed by overall sparsity. Simultaneously, we utilize a graph embedding module to effectively process non-Euclidean data through a Graph Convolutional Network (GCN). These components in spatio-temporal module enhance Satformer's ability to capture and exploit the nonlinear and complex information present in traffic data. Furthermore, a transfer module is incorporated to disseminate global context information throughout the model.

Our contributions are as follows:

- We designed ASSIT, which adopts a multi-head self-attention structure. It can learn the correlation representation of traffic data at different spatial and temporal scales. We added a sparsity threshold to the attention matrix to efficiently process a large number of sparse inputs. By dynamically adjusting the threshold value, ASSIT adapts to the sparsity levels of various datasets, thereby enhancing the model's inference efficiency. Additionally, ASSIT allows the model to dynamically allocate computational resources to regions of interest, making the model operate more efficiently and enhancing its scalability.

- To process non-Euclidean structured data, we introduce a graph embedding module within each module via GCN. Since the graph embedding module learns the relationship between nodes and neighbors adequately, it can extract the local and global information of nodes from non-Euclidean structured data. It improves the ability of the model to extract nonlinear spatio-temporal correlation.

- We add a transfer module to the Satformer framework, which can blend and reshape the traffic representation learned by the previous modules, conveying a global temporal and spatio perspective, while also helping to strengthen the generalization ability of the model on different types of datasets

This paper is structured as follows. Section 2 surveys relevant research. Section 3 explains our proposed Satformer methodology. Section 4 presents experimental verification and comparisons. Section 5 makes a discussion and concludes this paper.

## 2 Related Works

We provide a review of the existing work on network traffic estimation. Existing traffic data estimation methods can be mainly divided into matrix completion based, tensor completion based and neural network based methods.

Matrix Completion (MC) methods have found widespread application in the estimation of traffic data. Some algorithms, such as the convex relaxation method based on minimum nuclear norm approximation [12] and matrix factorization-based methods [13], leverage the linear spatiotemporal characteristics of traffic data to infer missing values. However, these methods are often too simplistic, which can lead to inaccurate estimations when applied to large-scale traffic data.

As an extension of matrix completion, the goal of tensor completion aims to reconstruct low-rank tensors based on sparse observations of their entries. Several studies have adopted tensor completion, including recent works [14, 15, 16].To achieve higher accuracy in traffic data estimation, these works propose the use of tensor completion methods, which can more comprehensively capture spatio-temporal features in traffic data, effectively. A typical work of such a method is LTC [17], which leverages the strong local correlation of the data to identify and complete each subtensor with low rank. However, many traffic estimation algorithms based on tensor completion rely on CANDECOMP/PARAFAC (CP) or Tucker decompositions, commonly using inner products as interaction functions. This approach can often reduce estimation performance to some extent due to its limited ability to capture both linear and nonlinear correlations in traffic data.

In recent years, deep learning methods have shown notable advancements in traffic network analysis. Notably, research such as NTF [18] and [19] have explored the application of deep learning models, including Recurrent Neural Networks (RNNs), to achieve adaptive grouping and prediction of traffic tensors within large-scale networks. Noteworthy among these efforts is CoSTCo [20], which incorporates two convolutional layers to extract features from stacked embeddings, enhancing awareness of network dynamics through the acquisition of complex spatio-temporal features. Recent studies [21, 22] employ meta-learning and other algorithms, alongside attention mechanisms, to dynamically adapt to rapid changes in traffic patterns within the network. However, current deep learning models may focus more on global features, while neglecting the local and hidden spatio-temporal correlations in traffic data, which may lead to suboptimal estimation effects

## 3 Estimation Model: Satformer

### 3.1 System Model & Problem Definition

In satellite networks, inter-satellite traffic data can be modeled as a time-space matrix, which reflects the data volume to be transmitted between all node-node pairs over satellite networks. For the problem statement of traffic estimation over satellite networks, we introduce the following symbols: $N$: Number of satellites, $T$: Discrete time steps, we define the inter-satellite traffic matrix $\mathcal{Y} \in R^{I \times J \times T}$, where $\mathcal{Y}_{ijt}$ represents the data transmission from satellite $i$ to satellite $j$ at time step $t$. The $t$-th layer of this matrix represents a discrete time step.

Considering the influence of spatio distance and transmission delay in satellite networks, we can adjust the inter-satellite traffic by introducing a weight matrix. Let $\mathcal{W} \in R^{N \times N}$ be the weight matrix representing spatial distance and transmission delay, where $\mathcal{W}_{ij}$ denotes the weight from satellite $i$ to satellite $j$. Thus, the adjusted inter-satellite traffic data matrix can be represented as $\dot{\mathcal{X}} = \mathcal{Y} \odot \mathcal{W}$, where $\odot$ denotes element-wise multiplication. Taking into account these factors, the mathematical modeling of inter-satellite traffic data can be expressed as follows:

$$\dot{\mathcal{X}}_{ijt} = \mathcal{Y}_{ijt} \cdot \mathcal{W}_{ij} \tag{1}$$

where $i, j = 1, 2, \ldots, N$, and $t = 1, 2, \ldots, T$. This model considers the spatio distance and transmission delay between satellites, allowing the traffic data matrix to more accurately reflect the actual communication scenarios in the satellite networks. In the process of sampling and recovering inter-satellite traffic data, we begin by introducing the sampling matrix $S$, the sampled data $\mathcal{X}$, and the nonlinear estimation function $F$. The sampling process can be expressed using mathematical notation: $\mathcal{X} = \dot{\mathcal{X}} \odot S$.

This process retains elements in the inter-satellite traffic matrix $\dot{\mathcal{X}}$ where the corresponding positions in the sampling matrix $S$ are 1, while setting other positions to zero, resulting in the sampled data matrix $\mathcal{X}$. To recover complete traffic data from the sampled data, we introduce a nonlinear estimation function $F$. This function involves a complex nonlinear mapping to better estimate actual traffic data.

$$\tilde{\mathcal{X}} = F(\mathcal{X}) \qquad (2)$$

where $\mathcal{X}$ represents sampled data, and $\tilde{\mathcal{X}}$ is the recovered data obtained through the non-linear estimation function $F$.

## 3.2 Satformer Overview

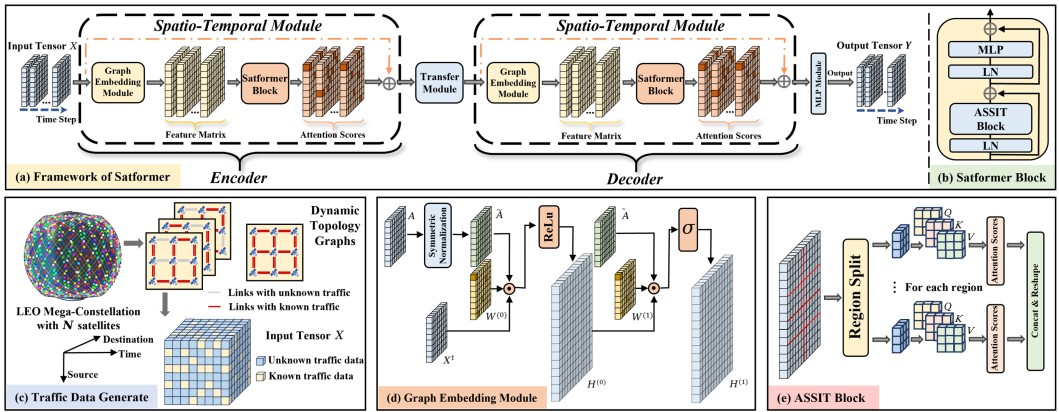

Figure 1: **(a)** Overall framework of our Satformer. **(b)** Details of a Satformer block. **(c)** Satellite network traffic data generation. **(d)** Details of a graph embedding module. **(e)** Details of an ASSIT block.

We design Satformer, a tensor completion model designed for the accurate and robust estimation of global traffic data in satellite networks. As illustrated in Fig. 1, Satformer is structured as an encoder-decoder architecture, with both components featuring multiple spatio-temporal modules. Residual connections interlink these modules to prevent neural network degradation. Each spatio-temporal module comprises a Graph Embedding Module and a Satformer Block. The key of Satformer to improve the estimation accuracy is that it can extract features efficiently and accurately from a large number of sparse satellite network traffic data. This is achieved through adaptive sparse spatio-temporal attention inside each Satformer block, facilitating the estimation of traffic data. A transfer module facilitates the seamless transmission of features from the encoder to the decoder. The encoder encodes the input traffic information, while the decoder is tasked with estimating the missing traffic data. The subsequent section provides a detailed description of each module.

## 3.3 Spatio-Temporal Module

Satformer utilizes spatio-temporal modules to extract spatio-temporal features from input tensors; this module primarily consists of graph embedding components and Satformer blocks.

**Graph Embedding**: The Spatio-Temporal module serves the goal of extracting spatio-temporal features from input tensor. Considering the inherent high sparsity of observed traffic data in real-world, it becomes imperative to represent tensors as low-dimensional vectors. Through the learning of embedded representations for nodes, the model inherently captures both structural and semantic information of nodes within the graph. This capability enables the model to comprehend relationships between nodes more effectively, facilitating the extraction of meaningful features from the $\mathcal{X} \in R^{I \times J \times T}$. Each Origin-Destination (OD) pair corresponds to an origin node, a destination node, and the traffic of the OD pair. To address the non-Euclidean nature of the data, particularly the spatio relationships within each OD pair, we employ Graph Embedding through Graph convolutional neural network (GCN), which has been widely used in many works [23, 24]. This approach allows the model to effectively handle non-Euclidean data, enhancing its capacity to capture and utilize the structural information present in the tensor $\mathcal{X} \in R^{I \times J \times T}$.

In Satformer, each Spatio-Temporal module contains a GCN model. A GCN model contains two layers of convolutional layer, the feature propagation rule can be stated as follows:

$$H^{(l+1)} = \sigma(\tilde{D}^{-\frac{1}{2}}\tilde{A}\tilde{D}^{-\frac{1}{2}}H^{(l)}W^{(l)}) \tag{3}$$

$$\tilde{A} = A + I, \tilde{D} = \sum_j \tilde{A}_{ij} \tag{4}$$

$$\mathcal{Z} = f(\mathcal{X}, A) = \sigma(\tilde{D}^{-\frac{1}{2}}\tilde{A}\tilde{D}^{-\frac{1}{2}}\text{ReLu}(\tilde{D}^{-\frac{1}{2}}\tilde{A}\tilde{D}^{-\frac{1}{2}}XW^{(0)})W^{(1)}) \tag{5}$$

where, $H^{(l)}$ signifies the node embedding matrix for layer $l$, $A$ represents the adjacency matrix and $I$ represents the self-connections matrix of $A$. $\tilde{A} \in R^{I \times J}$ represents the adjacency matrix with self-connections. $\tilde{D} \in R^{I \times I}$ denotes the degree matrix, which is a diagonal matrix with each element on the diagonal representing the sum of the corresponding row in $\tilde{A}$. The weight matrix for layer $l$ is denoted as $W^{(l)} \in R^{I \times M}$, and $H^{(l+1)} \in R^{I \times J \times M}$ represents the node embedding matrix for layer $l + 1$. $W^{(0)} \in R^{K \times L}$ denotes the weight matrix from the input layer to the hidden layer, and $W^{(1)} \in R^{K \times L}$ denotes the weight matrix from the hidden layer to the output layer. Both $\sigma(\cdot)$ and *ReLU* are activation functions employed in the model. $Z \in R^{I \times M \times K}$ represents the output embedding tensor.

**Satformer Block**: As shown in Fig. 1 (b), in each Satformer block, we use a layer normalization at the beginning to normalize the input embedding tensor. We then apply an ASSIT mechanism and a 2-layer MLP module for sparse spatio-temporal feature modeling and per-location embedding, respectively.

In the domain of communication network tensor completion, the spatio-temporal relationships among traffic data are complex, and it is necessary to model these relationships effectively. Traditional attention mechanisms, with their intensive nature, may encounter challenges related to high computational complexity and difficulties in capturing global relationships in such intricate scenarios. Several works proposed different sparse attention mechanism to mitigate such issue either relay on static patterns or skip computations in specific regions. As shown in Fig. 1(e), in this work, we explore an adaptive, sparse spatio-temproal mechanism. Detailed descriptions are as follows:

Given an input embedding feature tensor $\mathcal{Z} \in R^{I \times M \times K}$. First we divide the tensor slice into several local regions $\mathcal{Z}_{div}$, each of which has a size of $D \times D$. In our module, $\boldsymbol{Q}$, $\boldsymbol{K}$ and $\boldsymbol{V}$ respectively represent query, key, and value, which are used to calculate attention weights and generate the final output[25]. Then we calculate $\boldsymbol{Q}$, $\boldsymbol{K}$ and $\boldsymbol{V}$ tensor with linear projections for each region:

$$\boldsymbol{Q} = \mathcal{Z}_{div}W^q \quad \boldsymbol{K} = \mathcal{Z}_{div}W^k \quad \boldsymbol{V} = \mathcal{Z}_{div}W^v \tag{6}$$

where $W^q$, $W^k$ and $W^v$ are projection weights for query $\boldsymbol{Q}$, key $\boldsymbol{K}$ and value $\boldsymbol{V}$ respectively. We then consider introducing a local attention mechanism when calculating the attention score $\boldsymbol{\alpha}$ to make the model pay more attention to each local region in the input tensor. This improvement is designed to sharpen the model's attention specifically on local regions within the input sequence. The goal is to augment the expressiveness and robustness of the model by enabling it to capture and leverage more nuanced details and patterns present in localized segments of the input data. The implementation involves incorporating a position-related weight when calculating attention scores. The local attention in each region is operationalized through the use of a two-dimensional mask matrix $\Psi \in R^{D \times D}$, wherein, elements inside a defined center window $H$ are retained, while elements in other positions are set to zero. The size of the center window $H$ is a hyperparameter of the model, and its optimal value is determined through experiments on different datasets. The calculation of attention score $\boldsymbol{\alpha}_i$ for each region can be denoted as follows:

$$\boldsymbol{\alpha_i} = softmax(\frac{QK^T}{\sqrt{C}} \odot \Psi) \tag{7}$$

$$\boldsymbol{\alpha s_i} = softmax(W^s\text{ReLU}(1 - W^r\boldsymbol{\alpha_i}) \odot \boldsymbol{V}) \tag{8}$$

where $\odot$ is an element-wise product and $C$ is scaling factor. And the final output $\alpha_t$ are computed as:

$$\boldsymbol{\alpha_t} = concat(\boldsymbol{\alpha s_i}) \tag{9}$$

To regulate the sparsity of the attention scores and channel the model's focus onto specific portions of the input, an adaptive sparse regularization term is introduced. This involves applying L1 regularization to each element of the attention score matrix. The utilization of ReLU operations ensures that the

attention scores remain non-negative. Thus the sparse mask can be denoted as: $\text{ReLU}(1 - W^r \boldsymbol{\alpha_i})$. Finally, we apply the weighted Value to the attention score, while introducing additional learnable parameters to allow the model to adaptively learn the weighted sum of each position, resulting in the final output $\boldsymbol{\alpha_t}$, as shown in Eq. 9. Where, $W^r$ is the weighted matrix of L1 regularization, $W^s$ is the scaling matrix, both $W^r$ and $W^s$ are trainable parameters.

### 3.4 Transfer Module

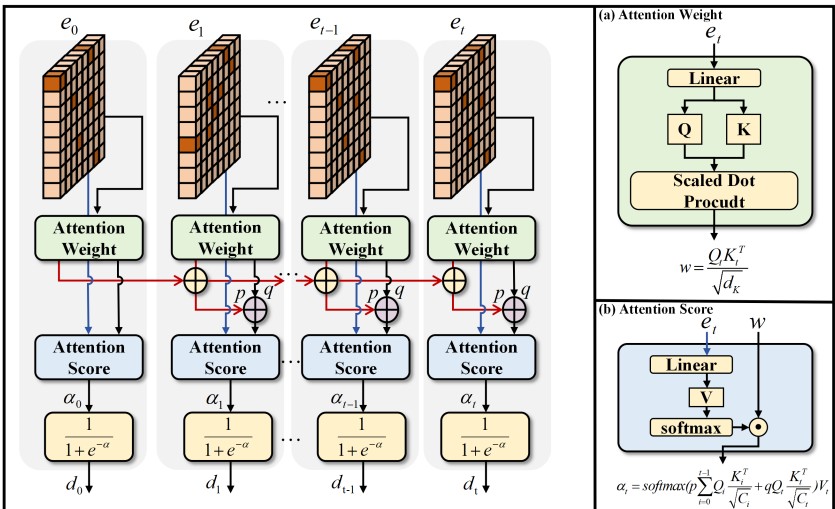

Figure 2: **Left** Details of transfer module. **(a)** Attention weight. **(b)** Attention score.

The conventional information transfer between the encoder and decoder typically relies on the output of the last layer of encoder. However, this approach may fall short in adequately conveying global context information, particularly when dealing with input tensors spanning a large number of time slices. The accumulation of errors over time can become a challenge. Consequently, it is necessary to add a module between encoder and decoder to effectively transfer the information. Satformer incorporates a self-attention-based transfer module between the encoder and the decoder. This module leverages Self-Attention, enabling the seamless transfer of globally contextual information learned in the encoder to the decoder. This augmentation empowers the decoder to more comprehensively consider information from the entire input sequence when generating output for each time slice, thus enhancing the estimation accuracy of missing values. Moreover, the transfer module enables the model to integrate spatio-temporal information in a more fine-grained manner, improving its adaptability to patterns across different temporal and spatio scales. The mathematical description of the Transfer Module is as follows:

Suppose the encoder outputs an eigenvector $e_t$ for each time step $t$ input $x_t$, where $t = 1, 2, \ldots, T$, then the output sequence of the encoder is $E = e_1, e_2, \ldots, e_T$. The goal of Transfer Module is to convert the output of the encoder $E$ to a new set of feature vector $D = d_1, d_2, \ldots, d_T$, where each $d_t$ is a feature enhanced representation corresponding to time step $t$. This process is achieved through the following self-attention mechanisms:

- Calculate Query, Key, and Value: The query vector $Q = EW^Q$ represents the query of future time points against past time points. The key vector $K = EW^K$ represents the encoding of a past point in time. The value vector $V = EW^V$ represents the specific characteristics of past time points. $W^Q$, $W^K$, and $W^V$ are learnable weight matrices.

- Calculate attention weight: Calculate the attention weight $\alpha_{t,t-1}$ of each time step $t$ and consider the effect that past attention scores exert on the present. $C_t$ is scaling factor at time step $t$, $i$ belongs to 1 to $t - 1$, $p$ and $q$ are parameters to control the effort of past time attention scores.

$$\alpha_{t,t-1} = softmax(p \sum_{i=1}^{t-1} Q_i \frac{K_i^T}{\sqrt{C_i}} + q Q_t \frac{K_t^T}{\sqrt{C_t}}) V_t \qquad (10)$$

- Generate transformation feature vector: Based on weights, attention to each time step $t$ to generate a new feature vector $d_t$.

$$d_t = \frac{1}{1 + e^{-\alpha_{t,t-1}}} \tag{11}$$

This process enables the Transfer Module to accurately measure the relationship between each future time point and all past time points, and to generate a new set of features that represent valuable information for future predictions.

## 3.5 Loss Function

During the training stage, the primary objective is to minimize the discrepancy between the actual and predicted traffic data. To achieve this, the loss function employed by Satformer is the mean square error (MSE), as expressed in Eq. 12. Additionally, to curtail the growth of model weights and mitigate the risk of overfitting, a penalty term is incorporated into the loss function.

$$L(\theta) = \frac{1}{|\bar{A}|} \sum_{(i,j,t)\in\bar{A}} \left( \chi_{ijt} - \tilde{\chi}_{ijt} \right) + \lambda \sum_i \left( \theta_i \right) \tag{12}$$

where $\bar{A}$ denotes the set of observed traffic data, $\chi_{ijk}$ and $\hat{\chi}_{ijk}$ are the truth and estimated traffic data respectively, $\theta$ represents all trainable parameters in Satformer, $\lambda$ is weight decay coefficient.

# 4 Experiments

## 4.1 Experimental Settings

**Datasets.** To assess the performance of Satformer, we employ it on three real-world satellite networks: Iridium, Telesat, and Starlink, thereby evaluating its capabilities across varying network scales: small-scale, medium-scale, and large-scale environments. Given the ongoing construction and utilization of many satellite networks, acquiring actual traffic data proves to be challenging. Thus, we generate corresponding traffic datasets using real satellite parameters and ground station coordinates. Similar methods have been used in many previous studies, and the specific details of this process are explained in the Appendix A. The traffic data collection interval was 1 second for all three datasets.

- **Iridium [26]**: The Iridium constellation comprising a total of 66 satellites uniformly distributed across 6 orbital planes. For our experimentation, we focus on the initial six periods, encompassing 36,000 time slices.
- **Telesat [27]**: It collects traffic data from the Telesat constellation which has a total of 298 satellites distributed in 26 orbital planes. We select the first five periods about 31500 time slots in our experiment.
- **Starlink [28]**: The traffic data recording originates from the Starlink constellation, comprising 1584 Low Earth Orbit (LEO) satellites evenly dispersed across 72 orbital planes. The first six periods about 32400 time intervals in our experiment.

For all three datasets, we divided the original dataset into a training set and a test set in an 8:2 ratio using the time slice partitioning method. We then used the training set for model training and the validation set for model validation and tuning. Subsequently, we constructed the test set by randomly masking portions of the training and validation sets that were not used for training. This approach ensures that the model is trained and validated on distinct segments of the data, which can help prevent overfitting and improve the model's ability to generalize to new, unseen data.

**Baselines.** For comparative analysis against our Satformer model, we select the following baseline models: three mathematical tensor completion models, namely HaLRTC, LATC and LETC, and four state-of-the-art neural network-based tensor completion models, CoSTCo, DAIN, SPIN and STCAGCN.

- **HaLRTC [29]**: A prototypical high-accuracy low-rank tensor completion algorithm utilizes the Alternating Direction Method of Multipliers (ADMMs) to attain precise outcomes, effectively managing dependencies among various constraints.

- **LATC [30]**: It introduces a novel regularization term, integrating temporal variation, into a third-order tensor completion model.
- **LETC [31]**: a Laplacian enhanced low-rank tensor completion framework for large-scale traffic speed kriging.
- **CoSTCo [20]**: An innovative Convolutional Neural Network (CNN)-based model developed for tensor completion to overcome the limitations associated with traditional low-rank tensor factorization approaches.
- **CDSA [32]**: A novel cross-dimensional self-attention approach for imputing missing values in multivariate, geo-tagged time series data.
- **DAIN [33]**: This method explicitly crafted to enhance the accuracy of neural tensor completion methods when predicting missing values within sparse, multi-dimensional datasets.
- **SPIN [34]**: An attention-based architecture using spatiotemporal graphs and autoregressive models for effectively reconstructing missing data in sparse, multivariate time series.
- **SAITS [35]**: a self-attention-based method for multivariate time series imputation that uses joint-optimization and diagonally-masked self-attention blocks.
- **STCAGCN [36]**: A graph-based deep learning method for traffic volume estimation by utilizing a graph attention-based speed pattern-adaptive adjacency matrix and a customized temporal attention mechanism.

**Evaluation Metrics.** Two widely employed metrics are applied to evaluate the estimation performance of Satformer. The calculation equations for these metrics are presented as follows:

- **Normalized Mean Absolute Error (NMAE)**:

$$NMAE = \frac{\sum_{(i,j,t)\in\bar{A}} |\chi_{ijt} - \tilde{\chi}_{ijt}|}{\sum_{(i,j,t)\in\bar{A}} |\chi_{ijt}|} \tag{13}$$

- **Normalized Root Mean Squared Error (NRMSE)**:

$$NRMSE = \sqrt{\frac{\sum_{(i,j,t)\in\bar{A}} |\chi_{ijt} - \tilde{\chi}_{ijt}|^2}{\sum_{(i,j,t)\in\bar{A}} \chi_{ijt}^2}} \tag{14}$$

where $\chi_{ijk}$ and $\tilde{\chi}_{ijk}$ represent the truth value and estimated value, $\bar{A}$ denotes the set of unobserved traffic data. **For both two metrics, the smaller they get to 0, the better the estimation performance of the model.**

### 4.2 Performance Comparison with Baselines

**Compare Satformer with mathematical baselines.** Table 1 provides a summary of the experimental results for our Satformer and the mathematical tensor completion baselines, HaLRTC, LATC and LETC. Performance evaluations, measured by NMAE and NRMSE, are conducted across three datasets with sampling ratios ranging from 2% to 10%. Our Satformer consistently outperforms the mathematical tensor completion algorithms, achieving significant improvements. Notably, even at the minimal 2% sampling ratio, Satformer maintains proficient performance, with NMAE values recorded as 0.098, 0.1017, and 0.1402 for the Iridium, Telesat, and Starlink datasets, respectively. In comparison, the leading mathematical models exhibit higher NMAE values of 0.2782, 0.2723, and 0.3784 under the same 2% sampling ratio. The observed performance enhancement in Satformer quantifies at 84.38%, 86.43%, and 106.77% for the respective datasets. Similar trends are also observed in NRMSE. These results indicate that mathematical models based on Alternating Direction Method of Multipliers or reliant on strong assumptions struggle to capture the complex spatio-temporal characteristics. In contrast, neural network-based models such as Satformer demonstrate formidable nonlinear representation capabilities, enabling effective extraction of spatio-temporal features from traffic data.

**Compare Satformer with neural network-based baselines.** Our Satformer outperforms the neural network-based baselines (CoSTCo, DAIN, SPIN, and STCAGCN) across all datasets, achieving the best estimation performance, as shown in Table 1. Notably, even with a 2% traffic data sampling rate, Satformer demonstrates significant improvements compared to the best-performing neural network-based baselines. On the Iridium dataset (66 satellites), Satformer improves NMAE and NRMSE by 8.57% and 8.95%, respectively. As the size of the dataset increases, performance

Table 1: Estimation Performance of Satformer Compared with Baselines

| Models | NMAE on Iridium | | | | | NRMSE on Iridium | | | | |
|---|---|---|---|---|---|---|---|---|---|---|
| | 2% | 4% | 6% | 8% | 10% | 2% | 4% | 6% | 8% | 10% |
| HaLRTC | 0.2782 | 0.2252 | 0.2044 | 0.1935 | 0.1886 | 0.3926 | 0.3381 | 0.3074 | 0.2888 | 0.2778 |
| LATC | 0.581 | 0.5809 | 0.5809 | 0.5809 | 0.5808 | 0.6009 | 0.5998 | 0.5997 | 0.5997 | 0.5996 |
| LETC | 0.1807 | 0.1672 | 0.1545 | 0.1439 | 0.1354 | 0.2591 | 0.2384 | 0.2203 | 0.1984 | 0.1861 |
| Improve% | 84.38% | 71.83% | 61.61% | 60.06% | 66.54% | 116.82% | 100.84% | 90.24% | 79.71% | 84.44% |
| CoSTCo | 0.1629 | 0.1623 | 0.16 | 0.1588 | 0.1435 | 0.5664 | 0.5644 | 0.5646 | 0.5621 | 0.5574 |
| CDSA | 0.1616 | 0.1601 | 0.1599 | 0.1598 | 0.1120 | 0.6632 | 0.6058 | 0.5219 | 0.5249 | 0.5103 |
| DAIN | 0.1159 | 0.1156 | 0.1150 | 0.1144 | 0.1126 | 0.1435 | 0.142 | 0.1391 | 0.1377 | 0.127 |
| SPIN | 0.1206 | 0.1185 | 0.1175 | 0.1170 | 0.1158 | 0.1302 | 0.1310 | 0.1291 | 0.1229 | 0.1181 |
| SAITS | 0.1106 | 0.1078 | 0.1075 | 0.1073 | 0.1051 | 0.1203 | 0.1201 | 0.1201 | 0.1174 | 0.1161 |
| STCAGCN | 0.1064 | 0.1059 | 0.1058 | 0.1049 | 0.1046 | 0.1847 | 0.1622 | 0.1523 | 0.1435 | 0.1203 |
| **Satformer** | **0.098** | **0.0973** | **0.0956** | **0.0899** | **0.0813** | **0.1195** | **0.1187** | **0.1158** | **0.1104** | **0.1009** |
| Improve% | 8.57% | 8.84% | 10.67% | 16.69% | 28.67% | 8.95% | 10.36% | 11.49% | 11.32% | 17.05% |

| Models | NMAE on Telesat | | | | | NRMSE on Telesat | | | | |
|---|---|---|---|---|---|---|---|---|---|---|
| | 2% | 4% | 6% | 8% | 10% | 2% | 4% | 6% | 8% | 10% |
| HaLRTC | 0.2723 | 0.2723 | 0.259 | 0.2538 | 0.2267 | 0.5518 | 0.4402 | 0.421 | 0.3968 | 0.3632 |
| LATC | 0.6193 | 0.6181 | 0.6129 | 0.6031 | 0.6002 | 0.6367 | 0.6367 | 0.6367 | 0.6367 | 0.6368 |
| LETC | 0.1896 | 0.1794 | 0.1637 | 0.1583 | 0.1513 | 0.2946 | 0.2751 | 0.261 | 0.2635 | 0.2534 |
| Improve% | 86.43% | 79.4% | 68.58% | 60.71% | 66.99% | 58.27% | 50.49% | 47.62% | 50.92% | 52.46% |
| CoSTCo | 0.2256 | 0.2182 | 0.2013 | 0.1898 | 0.1864 | 0.6996 | 0.6716 | 0.6482 | 0.6033 | 0.5852 |
| CDSA | 0.2354 | 0.2218 | 0.1565 | 0.1916 | 0.1815 | 0.6712 | 0.6523 | 0.5449 | 0.5014 | 0.4987 |
| DAIN | 0.1387 | 0.1345 | 0.1328 | 0.1297 | 0.1211 | 0.2687 | 0.2679 | 0.2538 | 0.2499 | 0.2476 |
| SPIN | 0.1298 | 0.1286 | 0.1278 | 0.1274 | 0.1273 | 0.2378 | 0.2365 | 0.2353 | 0.2347 | 0.2344 |
| SAITS | 0.1267 | 0.1223 | 0.1218 | 0.1113 | 0.1112 | 0.2213 | 0.2207 | 0.2201 | 0.2109 | 0.2013 |
| STCAGCN | 0.1488 | 0.1474 | 0.1457 | 0.1412 | 0.1393 | 0.2198 | 0.2184 | 0.2173 | 0.2184 | 0.2170 |
| **Satformer** | **0.1017** | **0.1** | **0.0971** | **0.0985** | **0.0906** | **0.1862** | **0.1828** | **0.1768** | **0.1746** | **0.1662** |
| Improve% | 27.63% | 28.6% | 31.62% | 29.34% | 33.66% | 18.05% | 19.47% | 22.91% | 25.09% | 30.57% |

| Models | NMAE on Starlink | | | | | NRMSE on Starlink | | | | |
|---|---|---|---|---|---|---|---|---|---|---|
| | 2% | 4% | 6% | 8% | 10% | 2% | 4% | 6% | 8% | 10% |
| HaLRTC | 0.3784 | 0.3392 | 0.3116 | 0.282 | 0.2558 | 0.6148 | 0.4796 | 0.4398 | 0.4116 | 0.3778 |
| LATC | 0.5738 | 0.5733 | 0.5737 | 0.5437 | 0.5348 | 0.5984 | 0.5982 | 0.5937 | 0.5938 | 0.5928 |
| LETC | 0.2899 | 0.2803 | 0.272 | 0.2656 | 0.2546 | 0.4937 | 0.484 | 0.4722 | 0.4624 | 0.4571 |
| Improve% | 106.77% | 103.71% | 101.63% | 101.82% | 108.17% | 79.26% | 77.81% | 77.78% | 74.23% | 75.33% |
| CoSTCo | 0.2553 | 0.2479 | 0.2466 | 0.2462 | 0.2428 | 0.6635 | 0.6548 | 0.6531 | 0.6519 | 0.6498 |
| CDSA | 0.2567 | 0.2119 | 0.2198 | 0.1964 | 0.2047 | 0.7732 | 0.6034 | 0.6032 | 0.5987 | 0.5975 |
| DAIN | 0.237 | 0.2231 | 0.2346 | 0.2233 | 0.2172 | 0.431 | 0.4036 | 0.4114 | 0.4189 | 0.4186 |
| SPIN | 0.2398 | 0.2353 | 0.2353 | 0.2352 | 0.2216 | 0.3989 | 0.3961 | 0.3959 | 0.3942 | 0.3919 |
| SAITS | 0.2499 | 0.2498 | 0.2314 | 0.2291 | 0.2215 | 0.3471 | 0.3412 | 0.3401 | 0.3349 | 0.3312 |
| STCAGCN | 0.1944 | 0.1891 | 0.1802 | 0.1754 | 0.1685 | 0.3644 | 0.3611 | 0.3653 | 0.3644 | 0.3625 |
| **Satformer** | **0.1402** | **0.1376** | **0.1349** | **0.1316** | **0.1223** | **0.2754** | **0.2722** | **0.2656** | **0.2645** | **0.2607** |
| Improve% | 38.66% | 37.43% | 33.58% | 33.28% | 37.78% | 32.32% | 32.66% | 37.54% | 37.77% | 39.05% |

improvements continue and escalate. On the Telesat dataset (298 satellites), Satformer achieves improvements of 27.63% in NMAE and 18.05% in NRMSE. For the Starlink dataset (1584 satellites), Satformer exhibits even more substantial improvements, with NMAE and NRMSE increasing by 38.66% and 32.32%, respectively. These results highlight Satformer's effectiveness in handling large-scale datasets, suggesting potential deployment in real-world satellite networks. The limitations

of CoSTCo are evident due to its exclusive reliance on two-dimensional convolution for spatial feature extraction without explicitly modeling temporal features. DAIN falls short by not explicitly modeling interactions between entities, which limits information utilization, despite its combination of information for data augmentation. SPIN's ability to handle sparsity or irregularly sampled data might be limited, which could affect the accuracy of traffic estimation in satellite networks where data is often incomplete. STCAGCN captures time-asynchronous correlations may not fully account for the complex temporal dynamics in satellite network, leading to less accurate estimations. The architecture of STCAGCN cannot ensure the information learned at earlier stages is preserved and utilized in later stages. Although CDSA also utilizes the self-attention mechanism, its dimension-wise processing may limit its ability to capture complex interactions. RNN-based models of SAITS are generally inferior to Transformer architectures in terms of handling long-distance dependencies and efficiency. In contrast, Satformer excels by explicitly incorporating both spatial and temporal features within each module. The graph embedding captures nonlinear information, the Satformer module integrates the ASSIT, and the transfer module seamlessly transmits global contextual information. This comprehensive design enables Satformer to deliver exceptional performance in inter-satellite traffic data estimation, effectively addressing the challenges of large-scale, sparsely populated datasets.

## 5 Conclusion and Discussion

This paper proposes Satformer, a novel traffic data estimation algorithm for large-scale satellite networks, aiming at fast and accurate estimating global traffic matrix from partial sampling in a cost-effective manner. Motivated by this, we design a region-aware sparse spatio-temporal attention mechanism to concentrate on specific local regions of the input tensor, where the input tensor is embedded in a graph convolutional neural network. Thus, spatio-temporal features from the traffic matrix are effectively extracted with computational efficiency and robustness.

Extensive experiments with datasets of varying scales-small, medium, and large have shown that Satformer has significant advantages on both accuracy and efficiency for traffic estimation compared with baselines, particularly in larger networks. Moreover, we analyze the robustness of Satformer under different conditions and further verify the role of each module through ablation studies. The results demonstrate the potential of Satformer for deployment in actual systems.

Despite Satformer is effective adopted for traffic estimation, deep learning models for traffic estimation remain mostly black boxes. It is quite important to understand the reasons behind inferences in the satellite networking domain. In addition, although Satformer is cost-effective, it is necessary to further reduce its computational complexity, considering the limited computational resources of existing satellites.

Future works should prioritize enhancing computational efficiency. It is also important to explore interpretability and decision basis of our deep learning model for traffic estimation. For example, explanation techniques, such as a local interpretable model-agnostic explanation (LIME) [37], are able to make a visual analysis of the model and analyze the internal working mechanism from specific examples. Additional explanatory tools, such as feature importance analysis, will help users in understanding the model's workings.

## Acknowledgments and Disclosure of Funding

We thank Xiaoshan Yu for the inspiration we got from chatting with him, and Xingyu Liu, our seven anonymous reviewers, and audience at ICML and this conference for helpful comments and suggestions. This work was supported in part by the National Natural Science Foundation of China under Grant 61934002 and 62102302, the Fundamental Research Funds for the Central Universities under Grant XJSJ23088, the National Key Laboratory of Advanced Communication Networks under Grant FFX22641X007, BAX24641X002, and also supported by The Youth Innovation Team of Shaanxi Universities.

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

# A   Traffic Generator

To the best of our knowledge, there is no publicly available inter-satellite traffic data or literature describing the traffic distribution for any existing satellite networks. This includes long-established networks like Iridium, as well as those currently under development, such as Starlink. The characteristics of traffic load borne by satellite networks are intricate. Satellites predominantly communicate with terrestrial terminals via ground stations. Spatially, the distribution of ground stations is uneven due to factors such as topography, economic considerations, and geopolitical influences. In regions with extreme environmental conditions or economic underdevelopment, such as oceans, deserts, and polar areas, the received traffic is significantly lower compared to more favorable environments, contributing to an uneven spatial distribution of traffic in the satellite network. Furthermore, the global distribution of earth stations spans various time zones, resulting in non-stationary traffic generation at different times. This temporal variability leads to significant traffic variations among stations. Additionally, to ensure link quality between satellites and ground stations and to mitigate the impact of frequent satellite handoffs, ground stations must consider multiple factors, including elevation angle, service time, and signal strength, when selecting communication satellites. This complexity adds to the challenges associated with managing traffic in satellite networks.

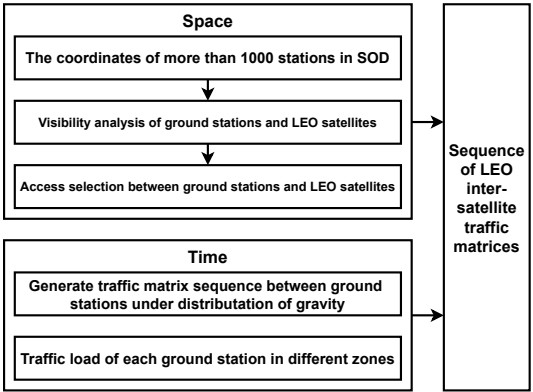

Figure 3: Traffic generation framework.

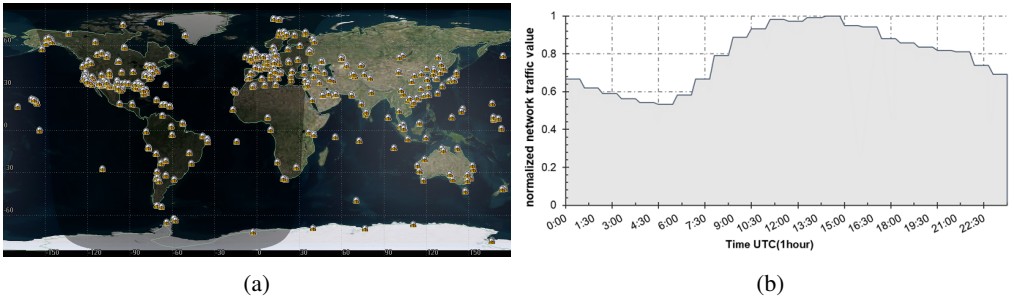

|     |     |
| :-: | :-: |
| (a) | (b) |

Figure 4: (a) Global distribution of satellite ground stations. (b) Normalized one-day traffic variation for ground station

The pivotal aspect in assessing the effectiveness of the proposed scheme is constructing a coherent satellite network traffic model that accurately represents the traffic characteristics of the satellite network. The devised traffic generation method, as illustrated in the accompanying figure, takes into account the spatial distribution of ground stations, temporal characteristics, and satellite-ground access. Although this approach primarily focuses on inter-satellite communication and excludes satellite-ground communication, it offers comprehensive consideration of three key factors: spatial distribution of ground stations, temporal dynamics, and satellite-ground access. The culmination of these considerations results in the generation of a sequence of inter-satellite traffic matrices.

**spatio distribution of ground stations.** Due to limitations imposed by antenna size, equipment volume, and quality, current user communications with satellites predominantly occur through ground

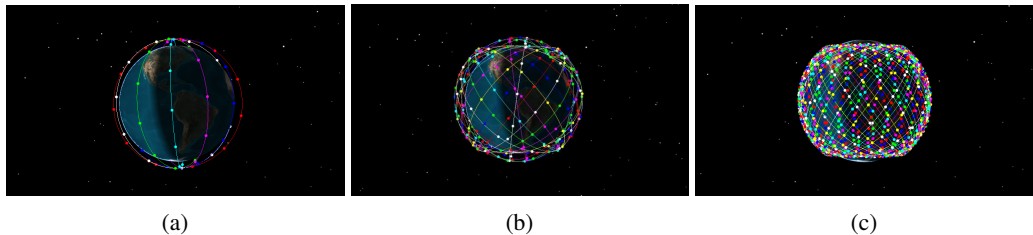

Figure 5: (a)Iridium constellation. (b)Telesat constellation. (c)Starlink constellation

stations. The spatial distribution of traffic load in a satellite network is intricately linked to the global geographical distribution of earth stations. In this work, ground stations are strategically positioned based on the coordinates provided by the Standard Object Database (SOD) within the Satellite Tool Kit (STK). The SOD database contains geographical location information for 1,016 Earth stations worldwide, primarily situated in islands, mountainous areas, rural locales, and other remote regions effectively served by satellites. In contrast to alternative assumptions regarding user distribution, leveraging the SOD ensures a more accurate representation of user distribution and density, which is crucial for this work. Consequently, this facilitates an effective depiction of the spatial distribution of traffic load within the satellite network.

**temporal dynamics.** The temporal variations in traffic load within non-geostationary orbit satellite networks predominantly arise from two factors: the diurnal fluctuations induced by regional local times and the geographic variations in daily traffic patterns influenced by global time zones. In the foreseeable future, the satellite optical network is expected to handle a traffic load comparable to that of the ground network, either matching, proportionally scaling, or exhibiting similar patterns. To ensure accuracy and effectiveness in generating traffic scenarios, this paper employs the four-month average daily traffic change trend from the GEANT network to characterize daily traffic variations. Fig.4b depicts the normalized cumulative traffic load over a 24-hour period, with the highest peak value normalized to 1. Notably, the flow intensity peaks around 12 noon, gradually diminishes, and then experiences a subsequent rise around 5 AM the following day. In addressing geographical variations, for the sake of model simplicity, the local time of a ground station within its respective time zone is incremented by one hour for every 15 degrees of longitude eastward from Greenwich Mean Time. This adjustment is contingent on the specific time zone associated with each ground station.

**satellite-ground access.** The capability of a ground station to establish satellite-ground communication links with multiple satellites within a given time window is influenced by factors such as satellite density and the coverage area of an individual satellite. Different satellite-ground access methods introduce varying effects on the characteristics of traffic load. Utilizing visibility analysis outcomes obtained through the Satellite Tool Kit (STK) and considering conditions for establishing satellite-ground links, the service time offered by all satellites visible to a ground station in each time window is computed. The satellite offering the longest service time is then selected for access, allowing for the flexibility to choose alternative or custom-designed satellite-ground access methods as needed. The ground station selects the destination node, following a uniform distribution. The traffic density at the ground station is jointly determined by the local time within the time zone and the spatial distribution of ground stations across each time zone.

Assuming the network time is based on GMT (Greenwich Mean Time) and considering the generation interval $\Delta t(s)$ of the traffic matrix, the computation process for the traffic matrix sequence $\{F_{ter}^t | t \in N^*\}$ between ground stations is as follows:

For any time $t$, the total traffic $D_t$ sent by all ground stations in the whole network is calculated as Eq.15.

$$D_t = offerload \times B \times n_{ter} \tag{15}$$

where, $offerload$ is the network traffic load, $B$ is the maximum bandwidth of the inter-satellite link, and $n_{ter}$ is the total number of ground stations.

The latitude and longitude coordinates of any ground station $i$ is $(x_i, y_i)$, then the local time $t_m(h)$ of the time zone $m$ of any ground station $i$ can be calculated by Eq.16:

$$t_m = \left\lfloor \frac{t}{3600} \right\rfloor + \left\lfloor \frac{x_i}{15} \right\rfloor \tag{16}$$

Combined with the normalized cumulative traffic load in 24 hours in Fig.4b, the traffic intensity weight of the time zone $m$ where the ground station $i$ is located can be calculated:

$$w_m^t = \frac{w_{t_m}}{w_{total}}, 0 \le m \le 23 \& m \in N^* \tag{17}$$

where, $w_{total}$ is the total traffic load and $w_{t_m}$ is the load at time $t$ corresponding to time zone $m$.

The traffic $(f_m^t)_z$ that a ground station $i$ located in time zone $m$ needs to send at time $t$ can be calculated by Eq.18 as follows.

$$\left(f_m^t\right)_i = \frac{D_t \times w_m^t}{n_m} \tag{18}$$

where, $n_m$ denotes the number of ground stations in time zone $m$, and $\sum\limits_{m=0}^{23} n_m = n_{ter}$.

Since the destination node is selected by the ground station according to uniform distribution, the traffic sent from the ground station $i$ to the ground station $j$ at time $t$ can be calculated by Eq.19:

$$F(i, j)^t = U(0.1, 1) * \left(f_m^t\right)_i \tag{19}$$

where, $F(i, j)$ denotes the traffic from the source ground station $i$ to the destination ground station $j$, and $U(0.1, 1)$ represents the uniform distribution from 0.1 to 1.

By traversing all the ground stations in the network at each time interval, the traffic matrix sequence $\{F_{ter}^t | t \in N^*\}$ between the ground stations can be calculated. According to the inter-satellite visibility analysis results provided by STK, as shown in Fig.5, combined with the satellite-to-ground access method described previously, the inter-satellite traffic matrix sequence $\{F_{sat}^t t \in N^*\}$ can be obtained.

## B  Theoretical Analysis

**Lemma 1.** *Convergence of GCN Layer-wise Propagation.*

**Statement**: For a multi-layer GCN with layer-wise propagation defined as $H^{(l+1)} = \sigma(D^{-\frac{1}{2}} \tilde{A} D^{-\frac{1}{2}} H^{(l)} W^{(l)})$, the node embeddings $H^{(l)}$ converge as $l$ increases, under mild conditions on the activation function $\sigma$ and weight matrices $W^{(l)}$. In our work, **Lemma 1** supports the use of two-layer convolutional layers in GCN to effectively propagate features, thereby validating the capability of GCN to capture and utilize the structural information present in the input tensor.

*Proof.* Consider a GCN with $L$ layers. The layer-wise propagation is given by:

$$H^{(l+1)} = \sigma(D^{-\frac{1}{2}} \tilde{A} D^{-\frac{1}{2}} H^{(l)} W^{(l)})$$

where $H^{(l)} \in \mathbb{R}^{N \times d_l}$ is the node feature matrix at layer $l$, $D$ is the degree matrix, $\tilde{A} = A + I$ is the adjacency matrix with self-loops, and $W^{(l)} \in \mathbb{R}^{d_l \times d_{l+1}}$ is the weight matrix.

Assume the activation function $\sigma$ is Lipschitz continuous with Lipschitz constant $L_\sigma$, i.e., for all $x, y \in \mathbb{R}$,

$$|\sigma(x) - \sigma(y)| \le L_\sigma |x - y|$$

Assume the weight matrices $W^{(l)}$ are bounded, i.e., there exists a constant $M$ such that $\|W^{(l)}\| \le M$ for all Define the propagation operator $\Phi : \mathbb{R}^{N \times d_l} \to \mathbb{R}^{N \times d_{l+1}}$ as:

$$\Phi(H) = \sigma(D^{-\frac{1}{2}} \tilde{A} D^{-\frac{1}{2}} H W)$$

To show $\Phi$ is a contraction, consider two node feature matrices $H_1, H_2 \in \mathbb{R}^{N \times d_l}$. We need to show that:

$$\|\Phi(H_1) - \Phi(H_2)\| \le k\|H_1 - H_2\|$$

for some $0 \le k < 1$.

Compute the difference:

$$\|\Phi(H_1) - \Phi(H_2)\| = \|\sigma(D^{-\frac{1}{2}}\tilde{A}D^{-\frac{1}{2}}H_1W) - \sigma(D^{-\frac{1}{2}}\tilde{A}D^{-\frac{1}{2}}H_2W)\|$$

Using the Lipschitz continuity of $\sigma$:

$$\|\Phi(H_1) - \Phi(H_2)\| \le L_\sigma\|D^{-\frac{1}{2}}\tilde{A}D^{-\frac{1}{2}}(H_1 - H_2)W\|$$

Apply the sub-multiplicative property of norms:

$$\|\Phi(H_1) - \Phi(H_2)\| \le L_\sigma\|D^{-\frac{1}{2}}\|\|\tilde{A}\|\|D^{-\frac{1}{2}}\|\|H_1 - H_2\|\|W\|$$

Since $D$ and $\tilde{A}$ are derived from the graph structure and $\|W\|$ is bounded by $M$:

$$\|D^{-\frac{1}{2}}\|\|\tilde{A}\|\|D^{-\frac{1}{2}}\| \le \lambda_{\max}$$

where $\lambda_{\max}$ is the largest eigenvalue of the normalized adjacency matrix.

Combining these, we get:

$$\|\Phi(H_1) - \Phi(H_2)\| \le L_\sigma\lambda_{\max}M\|H_1 - H_2\|$$

For $\Phi$ to be a contraction, we need:

$$L_\sigma\lambda_{\max}M < 1$$

If $L_\sigma\lambda_{\max}M < 1$, then $\Phi$ is a contraction mapping. By the Banach fixed-point theorem, every contraction mapping on a complete metric space has a unique fixed point. Therefore, the node embeddings $H^{(l)}$ will converge to a fixed point as $l$ increases. $\square$

**Lemma 2.** *Stability of Attention Mechanism with Masking.*

**Statement**: The attention mechanism with a mask matrix focusing on the center region of the input data remains stable and does not degrade performance, provided the mask matrix is appropriately designed. **Lemma 2** supports the introduction of a local attention mechanism in the adaptive sparse spatio-temporal attention mechanism, enabling the model to better capture and utilize details and patterns in localized regions of the input tensor.

*Proof.* The mask matrix $M$ is designed such that $M_{ij} = 0$ for elements outside the central region and $M_{ij} = 1$ within the central region. This implies that only the attention scores corresponding to the central region are retained, effectively reducing the complexity of the attention mechanism by filtering out less relevant data. Mathematically, $M$ acts as a sparsity-inducing regularizer:

$$A'(X) = \text{softmax}(M \odot (XW_1W_2^\top X^\top))$$

By focusing on the central region, $M$ ensures that the attention mechanism does not overfit to peripheral noise, enhancing generalization.

The central region often contains the most informative parts of the data, as observed in empirical studies (e.g., Fig.9(a)).

By applying $M$, the attention mechanism $A'(X)$ prioritizes the computation of attention scores within this region, thus capturing critical local patterns:

$$A'(X)_{ij} = \frac{\exp((M \odot (XW_1W_2^\top X^\top))_{ij})}{\sum_k \exp((M \odot (XW_1W_2^\top X^\top))_{ik})}$$

This prioritization ensures that the most relevant features are emphasized, leading to improved prediction accuracy.

To show that the deviation introduced by $M$ is bounded, consider the difference between the original and masked attention mechanisms:

$$\Delta A = A(X) - A'(X)$$

Since $M_{ij} \in \{0, 1\}$, it acts as a binary mask, thus the modification is limited to setting some attention scores to zero while retaining the rest:

$$\|\Delta A\|_F = \|\text{softmax}(XW_1W_2^\top X^\top) - \text{softmax}(M \odot (XW_1W_2^\top X^\top))\|_F$$

Given that softmax is a Lipschitz continuous function with constant 1, the Frobenius norm $\|\Delta A\|_F$ is bounded by the norm of the difference in the inputs:

$$\|\Delta A\|_F \leq \|(1 - M) \odot (XW_1W_2^\top X^\top)\|_F$$

Since $M$ zeros out peripheral entries, the difference is confined to the less relevant regions, ensuring that the overall deviation remains controlled. □

## C    Implemenation Details

Table 2: Hyper Parameter Settings

| Dataset | Model | lr | epochs | batch size |
|---------|-------|-----|--------|------------|
| Iridium | CoSTCo | 0.001 | 100 | 64 |
| | DAIN | 0.0001 | 50 | 256 |
| | SPIN | 0.0008 | 50 | 32 |
| | STCAGCN | 0.0005 | 50 | 32 |
| | Satformer | 0.001 | 200 | 128 |
| Telesat | CoSTCo | 0.001 | 100 | 64 |
| | DAIN | 0.0001 | 100 | 256 |
| | SPIN | 0.0008 | 50 | 32 |
| | STCAGCN | 0.0005 | 100 | 32 |
| | Satformer | 0.001 | 200 | 128 |
| Starlink | CoSTCo | 0.001 | 100 | 64 |
| | DAIN | 0.0001 | 50 | 1024 |
| | SPIN | 0.0008 | 100 | 64 |
| | STCAGCN | 0.0005 | 150 | 32 |
| | Satformer | 0.001 | 300 | 128 |

Satformer and the neural network-based baselines are implemented in PyTorch, while the mathematical baselines are implemented using Numpy. We evaluated Satformer against the baselines on a server equipped with an NVIDIA RTX 2080Ti GPU, 128 GB DDR4 RAM, and an Intel Xeon Silver 4208 CPU, running the Ubuntu 18.04 operating system. All models are trained for a range of 50 to 300 epochs with the first 5 epochs designated for warmup, and early stopping is adopted during the training process. The Adam optimizer [38] is used to optimize our model. A grid search strategy is applied to determine the best learning rate, epochs, and batch size. Based on the results of the grid search strategy, the optimal hyperparameters for the Satformer model and the neural network baselines are presented in Table 2, with the best weight decay determined to be 0.00001. We run each model 10 times with the same parameters and record the mean results in Table 1.

## D    Model Parameter Selection

**Impact of module number.** The module number indicates how many spatio-temporal modules should be contained in Satformer. It significantly impacts the computational efficiency and accuracy of Satformer. We adjust the number of modules from 1 to 20 and record the NMAE and NRMSE for each dataset, set the sample ratio to 2% and maintain the other hyperparameters constant. As shown in Fig.6, we observe that the error of Satformer continually decreases as the number of modules increases, up to a point (10 for all three datasets), after which it begins to increase. The reason is that increasing the number of modules enhances Satformer's ability to extract spatio-temporal features,

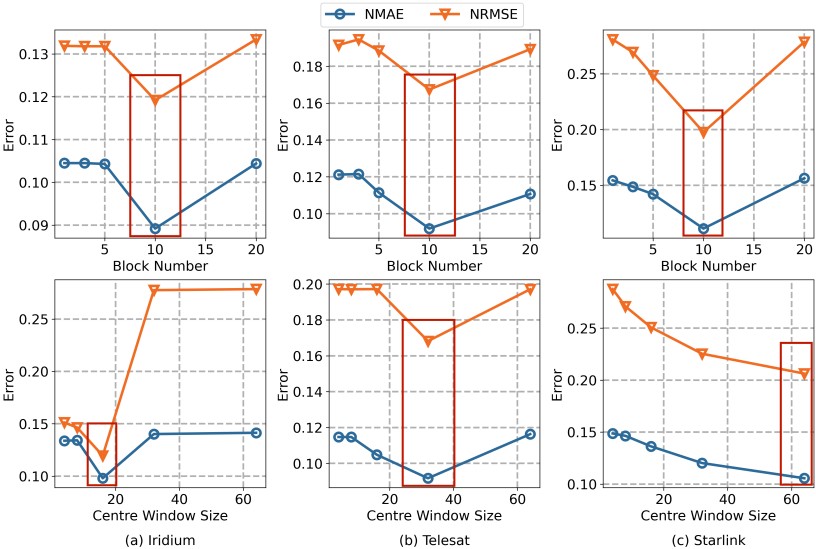

Figure 6: Analysis of hyper-parameters

while an excessive number of modules can lead to model overfitting and increased complexity. Consequently, we select a module number of 10 for all three datasets.

**Impact of centre window size.** The centre window size indicates how many elements should be retained in a mask matrix. It also significantly influences the accuracy of Satformer. It is another crucial hyperparameter of Satformer. The results in Fig. 6 show that, for Iridium and Telesat, the estimation performance of Satformer continues to improve until it starts to decrease at a certain point (16 for Iridium and 32 for Telesat). Conversely, for Starlink, the estimation performance continues to increase. It can be observed that the window size is proportional to the dataset's scale. For small and medium-scale datasets, the model may extract useful information with a small center window size, thereby improving computational efficiency. However, for large-scale datasets, a larger center window size enables the model to extract more features while maintaining high accuracy. Accordingly, we set the center window size to 16 for Iridium, 32 for Telesat, and 64 for Starlink.

# E  Robustness Analysis

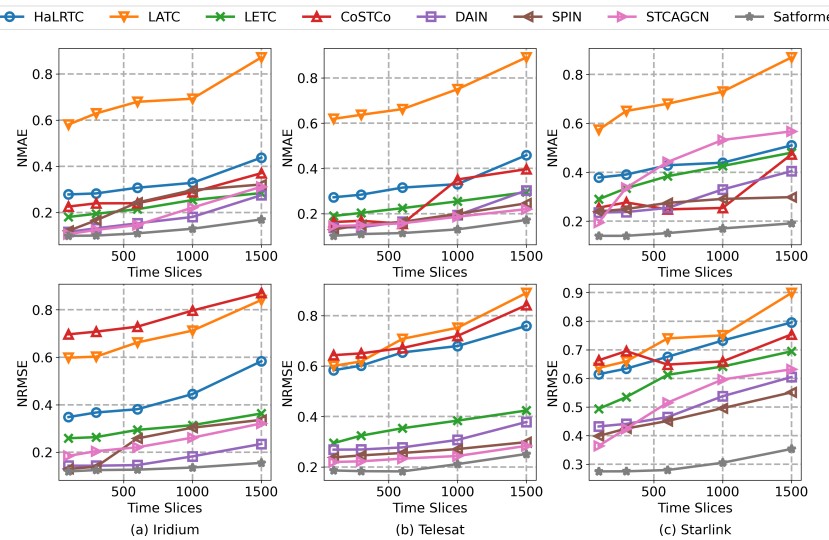

Figure 7: Analysis of robustness

To thoroughly evaluate the robustness and performance of Satformer, we conducted a comprehensive comparison of its Normalized Absolute Error (NMAE) and Normalized Root Mean Squared Error (NRMSE) metrics against all six baseline models across three diverse datasets. These datasets encompass varying numbers of time slices, ranging from 100 to 1500, providing a broad spectrum of temporal scales for analysis. As the number of time slices of the output increases, the accuracy will inevitably decrease, mainly because the complexity of the data increases exponentially with each additional dimension, making it harder to model and predict accurately [39]. In addition, In the process of tensor completion, especially over many time slices, there can be a loss of information that is critical for making accurate predictions. Each step in the prediction process may introduce slight inaccuracies, which accumulate over time [40]. The results from our experiments unequivocally demonstrate the superior and consistent performance of Satformer across all datasets, irrespective of the number of time slices involved. Notably, as the number of input time slices increases, Satformer consistently outperforms other models in terms of reliability, maintaining consistently low NMAE and NRMSE indicators. Importantly, even with the escalation of the temporal dimension and the expansion of dataset sizes, Satformer exhibits remarkable stability in its results. These findings underscore the robustness and scalability of Satformer, rendering it not only valuable in theoretical contexts but also highly applicable in real-world engineering scenarios.

## F  Ablation Study

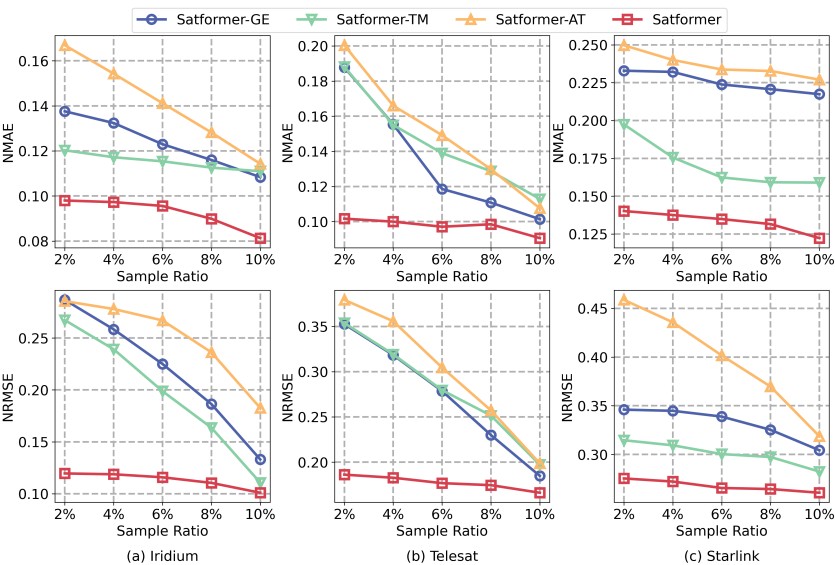

(a) Iridium
(b) Telesat
(c) Starlink

Figure 8: Ablation Study on Satformer

To assess the impact of our graph embedding module (GE), ASSIT module, and transfer module on Satformer's performance, we conducted ablation experiments, systematically removing these key components to create various model variants. Specifically, we evaluated the following variants: 1) Satformer-GE omits the graph embedding module; 2) Satformer-AT removes the adaptive sparse spatio-temporal attention mechanism module; 3) Satformer-TM does not incorporate the transfer module.

Experimental results consistently demonstrate the superior performance of the complete model as compared to the variant models from which crucial components have been removed, across three datasets that have varying sampling rates. Notably, the absence of the graph embedding module hinders the effective capture of topological structure information, underscoring its pivotal role in augmenting overall model performance. The significance of the ASSIT module becomes evident in addressing sparsity issues within spatio-temporal data; when it is absent, the model experiences a noticeable decline in performance, validating its effectiveness in uncovering temporal and spatial dependencies. Furthermore, the transfer module plays a crucial role in facilitating the conversion of information between different feature spaces, thereby enhancing the model's capability in feature representation.

## G    Runtime Analysis

Table 3: Training & Inference Time

| Model | Iridium | | Telesat | | Starlink | |
|---|---|---|---|---|---|---|
| | Training (s) | Inference (s) | Training (s) | Inference (s) | Training (s) | Inference (s) |
| HaLRTC | / | 123.9s | / | 520.6s | / | 2673.2s |
| LATC | / | 209.2s | / | 748.3s | / | 3350.1s |
| LETC | / | 62.4s | / | 253.4s | / | 1147.5s |
| CoSTCo | 164.6s | 0.200s | 570.4s | 0.274s | 1582.3s | **0.314s** |
| CDSA | 180.43s | 0.210s | 523.57s | 0.290s | 1478.37s | 0.308s |
| DAIN | 255.7s | 1.164s | 767.2s | 3.722s | 3254.8s | 13.475s |
| SPIN | 169.43s | 0.794s | 923.86s | 2.858s | 2566.5s | 3.291s |
| SAITS | 163.85s | 0.476s | 687.30s | 2.256s | 1879.57s | 3.433s |
| STCAGCN | 330.9s | 0.422s | 824.6s | 3.872s | 3993.8s | 3.8795s |
| Satformer | **80.3s** | **0.082s** | **168.9s** | **0.194s** | **879.5s** | 0.477s |

Table 3 presents a comparison of the training and inference times of Satformer with various baseline models. Although SPIN incorporates a sparse attention mechanism, it can be computationally intensive. This may lead to longer processing times. Notably, Satformer demonstrates the fastest training and inference times, which makes it particularly well-suited for real-world deployment. The significant reduction in these times is primarily attributed to the adaptive spatio-temporal attention mechanism, which introduces strategic sparsity in the sampling of input tensors and markedly reduces the number of parameters, offering a substantial time-saving advantage.

## H    Virtual Attention

The adaptive sparse spatio-temporal attention mechanism allows Satformer to focus on specific local regions of the input tensor, which is particularly beneficial for handling the large-scale and highly dynamic nature of satellite networks. We visualize the attention mechanism in Satformer to verify whether the functionality is achieved, as shown in Fig.9.

Fig.9(a) presents the initial traffic matrices for Iridium, Telesat, and Starlink satellite constellations. Each matrix's dimension is determined by the number of satellite nodes, denoted as $N$, with each point representing the volume of traffic between respective node pairs. Fig.9(b) illustrates the traffic matrices after applying a 10% sample rate to the three datasets, demonstrating a clear reduction in data density. Fig.9(c) displays the attention map, which are derived from the Satformer module using the attention scores, $\alpha_s$, as defined by Eq.9. The attention map's dimensions are $R$ ($D \times D \times$heads), which is then averaged across the head dimension and scaled to $N \times N$ in the $D \times D$ dimensions for visualization purposes. This scaling process is designed to be intuitive without altering the inherent relationships within the tensors. Finally, Fig.9(d) represents the estimated traffic matrices, which are the reconstructed traffic data from the sampled data.

Upon examining the initial traffic matrix for Iridium constellation, a notable volume of traffic is evident within the red-boxed area. After sampling, only sparse data points remain, yet the attention map successfully captures the significance of this high-traffic area, as indicated by the high attention scores. The estimated traffic matrix aligns well with the actual traffic in this region. Similar observations can be made for the Telesat and Starlink datasets, where the attention mechanism effectively identifies and emphasizes critical traffic areas, leading to accurate estimations within the reconstructed traffic matrices.

## I    General Tensor Completion Tasks

Although Satformer was developed to address traffic data estimation in satellite networks, the core strengths of its methodology endow it with the potential to be applied to other tensor completion

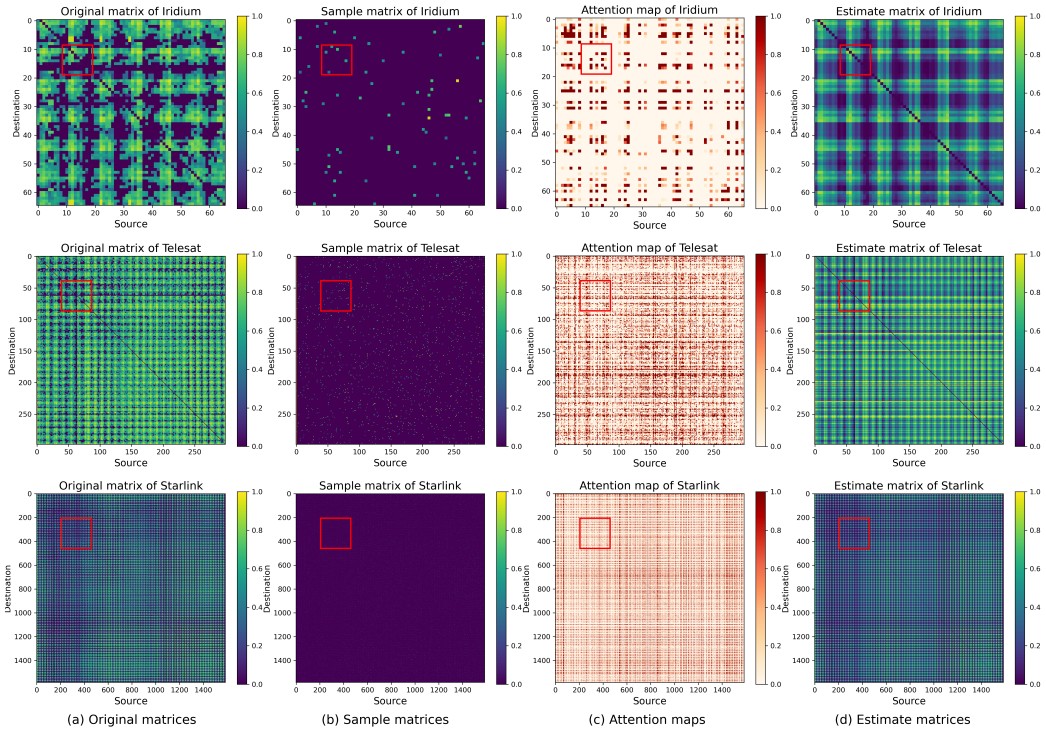

Figure 9: Visualization of adaptive sparse spatio-temporal attention mechanism

tasks requiring the handling of large-scale, sparse, and complex spatio-temporal characteristics. This includes, but is not limited to, social network analysis, environmental monitoring, bioinformatics, and other domains that necessitate the reconstruction and analysis of multidimensional data.

Table 4: Performance Under Foursquare tensor dataset

| Models | NMAE on Foursquare | | | | | NRMSE on Foursquare | | | | |
|---|---|---|---|---|---|---|---|---|---|---|
| | 2% | 4% | 6% | 8% | 10% | 2% | 4% | 6% | 8% | 10% |
| CoSTCo | 0.1465 | 0.1460 | 0.1454 | 0.1449 | 0.1438 | 0.2548 | 0.2513 | 0.2493 | 0.2425 | 0.2402 |
| DAIN | 0.1464 | 0.1460 | 0.1453 | 0.1450 | 0.1439 | 0.2434 | 0.2412 | 0.2401 | 0.2396 | 0.2368 |
| SPIN | 0.1322 | 0.1317 | 0.1308 | 0.1293 | **0.1291** | 0.1998 | 0.1973 | 0.1936 | 0.1922 | **0.1913** |
| STCAGCN | 0.1328 | 0.1321 | 0.1309 | 0.1296 | 0.1291 | 0.1999 | 0.1974 | 0.1933 | 0.1929 | 0.1918 |
| Satformer | **0.1320** | **0.1311** | **0.1304** | **0.1295** | 0.1294 | **0.1996** | **0.1967** | **0.1936** | **0.1920** | 0.1913 |

| Models | NRMSE on PeMS-Bay | | | | | NMAE on PeMS-Bay | | | | |
|---|---|---|---|---|---|---|---|---|---|---|
| | 2% | 4% | 6% | 8% | 10% | 2% | 4% | 6% | 8% | 10% |
| CoSTCo | 0.1513 | 0.1487 | 0.1461 | 0.1445 | 0.1432 | 0.2603 | 0.2499 | 0.2445 | 0.2400 | 0.2356 |
| DAIN | 0.1476 | 0.1462 | 0.1446 | 0.1430 | 0.1414 | 0.2447 | 0.2403 | 0.2358 | 0.2303 | 0.2249 |
| SPIN | 0.1357 | 0.1343 | 0.1329 | 0.1313 | 0.1299 | 0.2051 | 0.1997 | 0.1943 | 0.1889 | 0.1835 |
| STCAGCN | 0.1335 | 0.1321 | 0.1306 | 0.1290 | 0.1274 | 0.2012 | 0.1958 | 0.1904 | 0.1850 | 0.1796 |
| Satformer | **0.1308** | **0.1294** | **0.1280** | **0.1265** | **0.1249** | **0.1984** | **0.1930** | **0.1876** | **0.1822** | **0.1768** |

We use the Foursquare [33] and PeMS-Bay [41] tensor dataset to evaluate Satformer's performance on general tensor completion tasks. The Foursquare dataset is a point-of-interest tensor defined by user, location, and timestamp. Utilizing this dataset allows researchers and developers to gain insights into the dynamics of geographic social networks and to develop innovative applications and services

based on these insights. The PeMS-Bay data collect this traffic volume data from 325 loop sensors in the San Francisco bay area, ranging from January to March 2018 with a 5-min time interval.

The test results on two general datasets demonstrate the applicability of the Satformer method to other general tensor completion tasks. These positive results indicate that Satformer can serve as a powerful tool across various fields that involve complex spatio-temporal data, including social network analysis, traffic flow forecasting, environmental monitoring, and others. Naturally, additional adjustments and optimizations may be necessary for different application scenarios to achieve optimal performance.

## J Broader Impacts

Satformer is primarily used in emerging large-scale satellite communication networks. Currently, optimizing satellite network traffic engineering [6], [10] or topology engineering [11] relies on real-time collection of global traffic data. However, due to the limitations of satellite networks, real-time collection of comprehensive traffic status information incurs significant costs and overhead, making it nearly impossible to achieve, which further hinders the deployment of these solutions. Satformer addresses this issue by recovering global traffic data from a low sample rate (2%), which significantly reducing costs and overhead, thereby enabling the deployment of these solutions in real satellite networks. Additionally, Satformer helps improve network administrators' awareness of network states, optimizing network operations and maintenance. Beyond satellite communication networks, Satformer has the potential to be extended to other scenarios, such as transportation networks, the Internet of Things (IoT), and image processing. We have validated this in Appendix I. However, Satformer also presents interpretability risks. We recommend that researchers focus on interpretability when using Satformer, enhancing the transparency of application decisions related to traffic scheduling, routing, and congestion control.

