# OpenReview forum: "Satformer: Accurate and Robust Traffic Data Estimation for Satellite Networks"
_NeurIPS.cc/2024/Conference — NeurIPS 2024 poster_

### Official Review · Reviewer_6cUm · 2024-07-07

**Soundness:** 2
**Presentation:** 3
**Contribution:** 1
**Rating:** 3
**Confidence:** 4

**Summary:**

This paper proposed a satellite network traffic data estimation method which is called Satfomer.  The proposed method uses the adaptive sparse spatio-temporal attention mechanism to capture nonlinear spatio-temporal relationships. The proposed method is assessed on different satellite network datasets and compared to different existing methods.

**Strengths:**

1. The writing and organization of the paper are clear and easy to follow.
2. Using adaptive sparse spatio-temporal attention mechanism to capture nonlinear spatio-temporal relationships within satellite network traffic data seems feasible.
3. The proposed method is assessed on the satellite network traffic datasets of different scales and compared with existing methods.

**Weaknesses:**

1. The overall idea of the proposed is based on the attention mechanism, which is not novel in the field of spatio-temporal forecasting.
2. The motivation of this work is unclear, what is the main difference between the satellite network traffic data and other spatio-temporal data, e.g., traffic flow/speed data?
3. NMAE and NRMSE are used as the metrics to gauge the prediction performance of the methods in this work, what about MSE, MAE, and MAPE?
4. Some important baselines are missing, e.g., GraphWavenet, AGCRN, and GMAN, etc.

**Questions:**

See Weaknesses.

**Limitations:**

The novelty of the proposed method is limited.

---

> ### Author Rebuttal · Authors · 2024-08-03
>
> ## Dear Reviewer ZGQc:
> First and foremost, we would like to express our heartfelt gratitude for your diligent review of our research work and for providing valuable feedback. Your expertise and constructive comments are instrumental in enhancing the quality of our paper.
>
> Based on your comments, we have engaged in thorough discussions and made the necessary revisions. Below, we have addressed each of your points and outlined the corresponding modifications.
>
> To facilitate this discussion, we first retype your comments in italic font and then present our responses to the comments.
>
> ## Weakness 1：
> _The overall idea of the proposed is based on the attention mechanism, which is not novel in the field of spatio-temporal forecasting._
> ## Response W1:
>
> First, we appreciate your comment and sincerely apologize for any confusion caused by the ambiguous statements in our initial submission. We would like to make it clear that our research area is spatio-temporal traffic data estimation (imputation) , rather than traditional spatio-temporal forecasting. There are significant differences between the two. Spatio-temporal traffic data imputation mainly solves the problem of estimating global traffic data from partial sampling data in large-scale and dynamic satellite networks, while spatio-temporal prediction focuses more on predicting future data based on historical data. Our approach focuses on dealing with data sparsity and incompleteness inherent in satellite networks, improving data integrity and accuracy through efficient interpolation techniques.
>
> Second, we understand that you think the use of attention mechanisms is not novel, but we would like to further explain our specific innovation points on attention mechanisms to highlight the uniqueness and contribution of our work.
>
> ### 1. Adaptive Sparse Spatio-Temporal Attention Mechanism (ASSIT)
>
> #### i. Adaptive sparsity:
> The traditional attention mechanism is often inefficient when dealing with large-scale and sparse data. Our Adaptive Sparse Spatio-Temporal Attention mechanism (ASSIT) improves the computational efficiency and the ability to deal with sparse data by introducing a sparsity threshold and dynamically adjusting the sparsity of the attention matrix. This mechanism can automatically adjust the attention allocation according to the sparsity level of the input data, so that the model can still work efficiently in the case of limited computing resources.
>
> #### ii. Local area concerns:
> ASSIT specifically focuses on specific local regions of the input tensor, capturing details and patterns by applying higher attention weights to these regions. This approach not only enhances the ability of the model to capture complex spatio-temporal relationships, but also improves the robustness in the highly dynamic satellite network environment.
>
> ### 2. Graph embedding combined with attention mechanism
>
> #### i. Graph Convolutional Network (GCN) :
> We introduced GCN in each module to handle data with non-Euclidean structure. This combination enables the model to not only capture the spatial relationship between nodes, but also mine the spatio-temporal dynamic changes through the attention mechanism. This innovative combination makes the model perform well when dealing with complex spatio-temporal data such as satellite networks.
>
> #### ii. Fusion of global and local information:
> The transmission module transmits the global context information learned by the encoder to the decoder through the self-attention mechanism, so that the model can more comprehensively consider the information of the entire input sequence when generating the output. This effective fusion of global and local information significantly improves the model's ability to capture and represent spatio-temporal data.
>
> ### 3. Experimental validation and results
> Our experimental results show that Satformer significantly outperforms existing mathematical and neural network baseline methods when dealing with satellite network datasets of different sizes. The specific experimental results show that Satformer has a significant improvement in NMAE and NRMSE indicators, especially in large-scale networks, its advantage is more obvious.
>
> In summary, the innovation of our work in the attention mechanism is not only reflected in the improvement of the method, but also includes the breakthrough in the specific implementation way. We believe that these innovations can fully reflect the uniqueness and foresight of our research in the field of spatio-temporal traffic data interpolation.
>
> I hope the above explanation clarifies your doubts and recognizes the innovation and contribution of our work.

---

> ### Author Response · Authors · 2024-08-03
> **Continue**
>
> ## Weakness 2:
> _The motivation of this work is unclear, what is the main difference between the satellite network traffic data and other spatio-temporal data, e.g., traffic flow/speed data?_
>
> ## Response W2:
>
> We are grateful for your feedback. The dynamic shift of nodes, the variety of spatial distance and transmission delay, data sparsity, and incompleteness are the primary factors contributing to the complexity of satellite network traffic data, making its spatial characteristics more complex and challenging to model than those of other data sets (like transportation traffic data). A mathematical comparison of the intricacy of data from satellite networks is provided below:
>
> ### 1. Spatial distance and transmission latency
> In satellite networks, transmission delay and orbit altitude—both of which are dynamically changing—affect data transmission in addition to the physical distance between nodes. However, a traffic network can be represented by a relatively stable network diagram because the arrangement of the cars and roads is typically fixed or changes gradually.
>
> #### i. For the transportation network:
> The path length $d_{ij}$ is usually fixed, and the transmission delay is relatively stable, which is mainly affected by the traffic flow:
>
> $\tau_{ij}=f(d_{ij},v_{ij}(t))$
>
> where $v_{ij}(t)$ is the average speed at time $t$.
> ​
> #### ii. For satellite networks
> ​
> The transmission delay varies with time and is related to the orbital position of the satellite and the relative position between the ground station:
>
> $d_{ij}(t)=\sqrt{(x_i-x_j(t))^2+(y_i-y_j(t))^2+z_j(t)^2}$
>
> $\tau_{ij}(t)=f(\frac{d_{ij}(t)}c)+\delta(t)$
>
> where $(x_i, y_i) $ is the location of the ground station, $(x_j(t), y_j (t), z_j (t))$ $t$ is time the location of the satellite $d_{ij}(t)$ is the distance at time $t$, $c$ is the speed of light, $\delta(t)$ is the other delay factor (such as processing time).
>
> ### 2. Data sparsity and incompleteness
>
> Transportation traffic data usually come from fixed sensor networks or GPS devices, and these data sources are relatively stable in time and space. Although sparsity and incompleteness also exist, this sparsity has certain predictability and periodicity because traffic data usually exhibit strong temporal and spatial continuity.
>
> However, for satellite networks, due to the highly dynamic network topology, and the links between satellites and between satellites and ground stations may be intermittent for various reasons (e.g., geographical location, orbital position), resulting in sparse data. Most of the elements in the traffic data matrix $Y\in\mathbb{R}^{I\times J\times T}$ may be zero, since not all ground station-satellite and satellite-satellite pairs have traffic at each time step.
>
> At the same time, the traffic data in the satellite network changes rapidly and irregularly, and lacks the spatio-temporal continuity in the traffic data. And data missing is often unpredictable, as it can occur at any time and the spatio-temporal pattern of missing is not stable.
>
> Such sparsity and incompleteness make the flow data of satellite networks far more complex than other data sets such as traffic data, which requires higher-order algorithms and models for accurate analysis and estimation.

---

> ### Author Response · Authors · 2024-08-03
> **Continue**
>
> ## Weakness 3:
> _NMAE and NRMSE are used as the metrics to gauge the prediction performance of the methods in this work, what about MSE, MAE, and MAPE?_
>
> ## Response W3:
>
> Thank you for your comment. Although MSE, MAE and MAPE are also commonly used evaluation metrics, they may not be suitable to evaluate satellite network traffic estimation methods. The primary evaluation metrics used in this work, NMAE and NRMSE, as well as MSE, MAE and MAPE, are calculated by the following equations, where ${{\\chi _{ijt}}}$ and ${{\tilde {\\chi} _{ijt}}}$ represent the truth value and estimated value, ${\bar {\rm A}}$ denotes the set of unobserved traffic data.
>
> $NMAE=\\frac{ \\sum\\nolimits_{(i,j,t) \\in \\bar {\\rm A}} {| {{\\chi _{ijt}}} - {{\\tilde {\\chi} _{ijt}}} |} } {\\sum\\nolimits _ {(i,j,t) \\in \\bar {\\rm A}} {| {{\\chi _{ijt}}}|}}$
>
> $MAE=\\frac{ \\sum\\nolimits_{(i,j,t) \\in \\bar {\\rm A}} {| {{\\chi _{ijt}}} - {{\\tilde {\\chi} _{ijt}}} |} } {t}$
>
> $NRMSE=\\sqrt {\\frac{ \\sum\\nolimits_{(i,j,t) \\in \\bar {\\rm A}} {| {{\\chi _{ijt}}} - {{\\tilde {\\chi} _{ijt}}} |}^2 } {\\sum\\nolimits _ {(i,j,t) \\in \\bar {\\rm A}} {| {{\\chi _{ijt}}}|}^2} }$
>
> $MSE=\\frac{ \\sum\\nolimits_{(i,j,t) \\in \\bar {\\rm A}} {| {{\\chi _{ijt}}} - {{\\tilde {\\chi} _{ijt}}} |} ^ 2} {t}$
>
> It can be seen from the equations that NMAE and NRMSE are MAE and RMSE normalized by the truth value range, and RMSE is the root of MSE. Compared with MAE and MSE, it can not only effectively measure the estimation error, but also take into account the size of the error and its variability in different traffic data estimation scenarios in satellite networks.
>
> $MAPE=\\frac{  100\\% \\times \\sum\\nolimits_{(i,j,t) \\in \\bar {\\rm A}} {| \\frac { {{\\chi _{ijt}}} - {{\\tilde {\\chi} _{ijt}}} }{ {{\\chi _{ijt}}}}|} } {t}$
>
> From the equation of MAPE, it can be seen that its truth value cannot be 0; however, the truth data collected from the satellite networks may be 0, so it cannot be used as an evaluation metric in this work.
>
> ## Weakness 4:
> _Some important baselines are missing, e.g., GraphWavenet, AGCRN, and GMAN, etc._
> ## Response W4:
>
> Thank you very much for your comments. Our research objectives and application scenarios are fundamentally different from GraphWavenet, AGCRN, and GMAN. The main task of GraphWavenet and GMAN is to predict the traffic state at future moments, while our work focuses on interpolating the existing flow data so as to recover the complete flow data for further analysis and use. Although these two types of tasks both involve the processing of spatio-temporal data, the focus and specific application scenarios are different, so taking these models as baseline models is not suitable for our study. Here's a more detailed explanation:
>
> ### 1. GMAN [1]
> The core task of GMAN is spatio-temporal traffic prediction. It achieves the prediction of future traffic conditions by encoding and decoding historical traffic data. Mathematically, the basic flow includes:
>
> Input: Historical traffic data $X = (X_{t1}, X_{t2},... , X_{tP}) \in \mathbb{R}^{P \times N \times C}$, where P is the number of historical time steps, N is the number of sensors, and $C$ is the feature dimension.
>
> Output: predict the future traffic data $\hat{Y} = (\hat{X}_{t{P+1}}, \hat{X}_{t{P+2}}, ..., \hat{X}_{t{P+Q}}) \in \mathbb{R}^{Q \times N \times C}$, $Q$ is predicted time steps.
>
> ### 2. GraphWaveNet [2]
> Input: The input of GraphWaveNet model is a multi-dimensional spatio-temporal traffic data sequence and related graph structure information. These include:
>
> Historical traffic data series:
> Representation: $\mathbf{X} \in \mathbb{R}^{P \times N \times C}$
> Where $\mathbf{X}$ is a 3D tensor, $P$is the historical time step, $N$ is the number of sensors, and $C$ is the feature dimension (usually traffic flow, speed, etc.).
>
> Graph structure information:
> Adjacency matrix: $\mathbf{A} \in \mathbb{R}^{N \times N}$
> Description: $\mathbf{A}$ represents the adjacency relation of the sensor network, which is used to model the spatial dependencies.
>
> Output:
> The output of the GraphWaveNet model is the predicted value of the future traffic data. These include:
>
> Future traffic data series:
> Representation: $\hat{\mathbf{Y}} \in \mathbb{R}^{Q \times N \times C}$
>
> Where $\hat{\mathbf{Y}}$ is a 3D tensor, $Q$ is the time step to predict, $N$is the number of sensors, and $C$ is the feature dimension (typically traffic flow, speed, etc.).

---

> ### Author Response · Authors · 2024-08-03
> **Continue**
>
> ### 3. AGCRN [3]
>
> Input:
> The input of AGCRN includes historical time series data and dynamic graph structure. Specifically, the input data is of the following form:
>
> Historical time series data: denoted as $X = (X_{t1}, X_{t2},... , X_{tP}) \in \mathbb{R}^{P \times N \times C}$, where P is the number of historical time steps, N is the number of nodes, and $C$ is the number of features. Each $X_{ti}$ represents the graph node feature matrix at the $i$ time step.
>
> Dynamic graph structure: adaptive graph convolution is used to capture the dynamic spatial dependencies, and an adaptive adjacency matrix $\mathbf{A}_{\text{adaptive}}$ is used, which is learned from data and represents the time-varying relationship between nodes.
>
> Output: The output of AGCRN is the predicted future time series data, which is expressed as:
>
> $\hat{Y}=(\hat{X}_{t_{P+1}},\hat{X}_{t_{P+2}},... ,\hat{X}_{t_{P+Q}})\in\mathbb{R}^{Q\times N\times C}$
>
> where, $Q$ is the number of time steps to predict, and $\hat{X}_{t{P+1}}$ is the graph node feature matrix at the first future time step to predict.
> ### Our model (Satformer)
>
> Our research is not purely historical traffic data interpolation, but real-time interpolation and recovery for traffic data, which has significant differences in processing. Our core task is to deal with missing traffic data so that the data is more complete and reliable for further analysis. Our research objectives and methods include:
>
> Input: Traffic flow matrix with missing data $\mathbf{X} \in \mathbb{R}^{T \times N \times N}$, where $T$is the number of time steps and $N$is the number of satellites.
>
> Output: The complete padded flow $\mathbf{\hat{X}} \in \mathbb{R}^{T \times N \times N}$.
>
> Model structure: Interpolation model based on spatio-temporal graph neural network, using adjacency matrix and time series information for data recovery.
>
> In summary, our research goal is fundamentally different from GraphWavenet, AGCRN, and GMAN. While the former focuses on spatio-temporal traffic prediction, we focus on real-time interpolation and recovery of flow data to ensure data integrity and reliability. It is hoped that these detailed mathematical explanations and comparisons can help clarify how our study differs from other models.
>
> ### Comparison of other state-of-the-art models
>
> However, in order to further verify the effectiveness of our research results, we added other models (CDSA [4], SAITS [5]) for comparison, and the results are shown on the _TABLE II_ and _TABLE III_ in pdf at the **Author Rebuttal** Block highlighted in orange.
>
> Satformer is based on the Transformer architecture, which is very powerful in handling long-distance dependencies and suitable for handling time series data with variable patterns. Although CDSA also utilizes the self-attention mechanism, its dimension-wise processing may limit its ability to capture complex interactions. Rnn-based models of SAITS are generally inferior to Transformer architectures in terms of handling long-distance dependencies and efficiency.
>
> **Reference**
>
> [1] Zheng, Chuanpan, et al. "Gman: A graph multi-attention network for traffic prediction." Proceedings of the AAAI conference on artificial intelligence. Vol. 34. No. 01. 2020.
>
> [2] Wu, Zonghan, et al. "Graph wavenet for deep spatial-temporal graph modeling." arXiv preprint arXiv:1906.00121 (2019).
>
> [3] Gao, Xicai, et al. "An AGCRN Algorithm for Pressure Prediction in an Ultra-Long Mining Face in a Medium–Thick Coal Seam in the Northern Shaanxi Area, China." Applied Sciences 13.20 (2023): 11369.
>
> [4] Ma, Jiawei, et al. "CDSA: cross-dimensional self-attention for multivariate, geo-tagged time series imputation." arXiv preprint arXiv:1905.09904 (2019).
>
> [5] Du, Wenjie, David Côté, and Yan Liu. "Saits: Self-attention-based imputation for time series." Expert Systems with Applications 219 (2023): 119619.

---

> ### Author Response · Authors · 2024-08-03
> **Continue**
>
> ## Limitations 1:
> _The novelty of the proposed method is limited._
>
> ## Response L1:
>
> Thank you for your kindly comment and for recognizing the effort that went into our work. Your concern about novelty is valuable.
>
> First of all, satellite networks experience long link delays, high dynamics, and limited computing resources, resulting in high overhead and costs for measuring global traffic data. Thus, the primary focus of this paper is to reconstruct the remaining traffic data from a small sample (2%) of the satellite network data, rather than predicting future traffic. These traffic estimation methods encounter unique challenges due to the highly dynamic nature of satellite networks and the sparse and incomplete nature of traffic data. To the best of our knowledge, this is the first work on satellite network traffic estimation.
>
> Follow your suggestion, we restate the novelty. Our methodology, named "Satformer," differs from existing methods. We propose an Adaptive Sparse Spatio-Temporal Attention Mechanism (ASSIT), which dynamically adjusts the sparsity threshold of the attention matrix to enhance the model's inference efficiency when handling large-scale sparse inputs. This mechanism enables the model to adaptively allocate computing resources based on the sparsity level of different datasets, thereby improving operational efficiency and scalability.
>
> Additionally, a Graph Convolutional Network (GCN) is introduced as a graph embedding module to handle data with non-Euclidean structures. This module effectively captures local and global information between nodes, enhancing the model's ability to extract nonlinear spatio-temporal correlations.
>
> Furthermore, we introduce a transfer module to ensure no omissions occur in the long-term transmission between the encoder and decoder. This module employs a self-attention mechanism to comprehensively consider the entire input sequence when generating the output for each time slice, thereby improving the accuracy of missing value estimations.
>
> In summary, our work introduces innovations in several aspects, with its effectiveness and superiority verified through mathematical proofs and experiments.

---

> > ### Comment · Reviewer_RAyE · 2024-08-10
> >
> > Thanks for the informative and detailed reply. I suggest including these responses in the paper as appropriate.

---

> > > ### Author Response · Authors · 2024-08-10
> > > **Reply to Reviewer RAyE**
> > >
> > > Thank you for your suggestions, we will include these responses in the camera-ready version if this paper is accepted. Thank you again for your valuable comments.

---

> > ### Comment · Reviewer_6cUm · 2024-08-12
> > **reply to rebuttal**
> >
> > Thanks the authors for the comments. I would say the the main contributions of this paper are application-focused (satellite networks). Adaptive learning spatiotemporal correlations is not so novel in this field of  time series modelling, e.g., AGCRN and many other existing works (maybe with slight differences). Thus, the novelty of the paper is limited.
> >
> >
> > Moreover, in terms of spatiotemporal imputation, one could also consider exploring more advanced techniques, for example:
> > Liu, Mingzhe, Han Huang, Hao Feng, Leilei Sun, Bowen Du, and Yanjie Fu. "Pristi: A conditional diffusion framework for spatiotemporal imputation." In 2023 IEEE 39th International Conference on Data Engineering (ICDE), pp. 1927-1939. IEEE, 2023.

---

> > > ### Author Response · Authors · 2024-08-12
> > > **Reply to Reviewer 6cUM's Comments**
> > >
> > > Thank you for your reply. While we acknowledge that adaptive learning of spatio-temporal correlations has been applied to time series modeling in other fields, this does not mean that the novelty of our work is limited. Indeed, our main contributions focused on cost-effective traffic measurement over satellite networks.
> > >
> > > Different from terrestrial networks, satellite networks have unique challenges, such as spatial and temporal dynamics, sparsity and incompleteness, and limited computational resources. Our model, Satformer, effectively addresses these challenges through the ASSIT mechanism. This mechanism is specifically designed to focus on key data regions, enhancing the model's ability to capture complex, nonlinear spatiotemporal patterns essential for large-scale, highly dynamic environments like satellite networks. To the best of our knowledge, these challenges have not been sufficiently explored in existing literature, including the references you suggested.
> > >
> > > Satformer is theoretically convergent and stable, and extensive experiments demonstrate its accuracy and robustness with low overhead. This represents a significant contribution to the field, addressing the high overhead and costs associated with measuring global traffic data and meeting the urgent needs of satellite network management, operation, and maintenance. Unlike existing models such as AGCRN, Satformer specifically caters to the intricacies of satellite networks. In addition, the other three reviewers also acknowledged our novelty.
> > >
> > > The advanced method you recommended, Pristi, requires multiple inputs, including observed spatio-temporal data, geographic information, interpolation data, and noise information, which limits its application on satellite networks. In contrast, our method, Satformer, only requires observed spatio-temporal data as input, and has a lower time complexity of $O(NK)$ compared to Pristi's $O(Nkd)$.
> > >
> > > We initially intended to compare our method with Pristi, but since the author has closed the source code link and there is only one day left for discussion, it is impractical to reproduce Pristi. However, this does not impact the conclusion of our experiment, as our Satformer has been extensively compared with SOTA methods from 2021, 2022, and 2023 across multiple datasets.

---

### Official Review · Reviewer_78MM · 2024-07-12

**Soundness:** 3
**Presentation:** 3
**Contribution:** 3
**Rating:** 7
**Confidence:** 5

**Summary:**

The paper introduces Satformer, a novel method for accurate and robust traffic data estimation in satellite networks. It addresses the challenges of large-scale and dynamic nature of satellite networks by proposing an adaptive sparse spatio-temporal attention mechanism that focuses on specific local regions of the input tensor, enhancing the model's sensitivity to details and patterns. Satformer incorporates a graph convolutional network for processing non-Euclidean data and a transfer module for disseminating global context information, demonstrating superior performance over mathematical and neural baseline methods, especially for larger networks. Extensive experiments on simulated satellite network datasets, including Iridium, Telesat, and Starlink, show Satformer's significant improvements in reducing errors and maintaining robustness. The paper concludes that Satformer shows promise for deployment in actual systems and potential application in other tensor completion tasks requiring handling of complex spatio-temporal data.

**Strengths:**

1. The paper introduces Satformer, a neural network architecture that incorporates an adaptive sparse spatio-temporal attention mechanism (ASSIT) for estimating traffic data from partial measurements. ASSIT focuses on specific local regions of the input tensor to improve the model's sensitivity to details and patterns, enhancing its capability to capture nonlinear spatio-temporal relationships. The integration of a graph convolutional network (GCN) for processing non-Euclidean data and a transfer module for disseminating global context information are innovative aspects of the model.
2. The paper provides a detailed explanation of the system model, the proposed Satformer architecture, and the components within it, including the graph embedding module and the transfer module. The experiments are rigorous, conducted on datasets of varying scales, and compared against several baseline methods, demonstrating Satformer's superior performance.
3. The paper is well-structured, with a clear abstract, introduction, methodology, experimental setup, results, discussion, and conclusion. The figures and tables are used effectively to illustrate the model's architecture and the results of the experiments.
4. The work addresses a critical need for cost-effective traffic measurement methods in satellite networks, which is essential for network monitoring, routing, and performance diagnosis.

**Weaknesses:**

1. The paper does not present any theoretical results or proofs to support its claims, focusing instead on empirical results.
2. The paper primarily focuses on the application of Satformer in satellite networks. While it mentions potential applications in other areas, it does not provide extensive testing or results in those domains, which may limit the perceived generalizability of the model.
3. The paper may rely on certain assumptions that are not fully discussed or tested under various conditions. For example, the model's performance might vary under different network configurations or traffic patterns that were not part of the evaluation.

**Questions:**

1. What is the unique challenges of traffic data estimation for satellite networks, compared with other network types?
2. The research motivation is not convincing when there is few relevant literatures.

**Limitations:**

1. Only simulated datasets are used and it is difficult to collect real-world datasets to validate the proposed method.

---

> ### Author Rebuttal · Authors · 2024-08-03
>
> ## Dear Reviewer 8BTK:
> Initial and foremost, we would like to sincerely thank you for your thorough analysis of our study and insightful comments. Your knowledge and insightful criticism have been very helpful in improving our paper's quality.
>
> We have had in-depth talks and implemented the required changes in response to your feedback. We have addressed each of your concerns and described the appropriate changes below.
>
> To facilitate this discussion, we first retype your comments in italic font and then present our responses to the comments.
>
> ## Weakness 1:
> _The paper does not present any theoretical results or proofs to support its claims, focusing instead on empirical results._
> ## Response W1:
> Thank you for your constructive suggestions. For our work, we did favor experimental effects as support in the original paper. In our study, we do have some conclusions or citations to support our claims. **For a better presentation, We show them below and will add them to the appendix in the final version of this paper if it is accepted.**
>
> ### Lemma 1: Convergence of GCN Layer-wise Propagation
>
> **Statement:** For a multi-layer Graph Convolutional Network (GCN) with layer-wise propagation defined as
>
> $H^{(l+1)} = \sigma(D^{-\frac{1}{2}} \tilde{A} D^{-\frac{1}{2}} H^{(l)} W^{(l)})$
>
> the node embeddings $H^{(l)}$ converge as $l$ increases, under mild conditions on the activation function $\sigma$ and weight matrices $W^{(l)}$​.
>
> **In our work**:  Lemma 1 supports the use of two-layer convolutional layers in GCN to effectively propagate features, thereby validating the capability of GCN to capture and utilize the structural information present in the input tensor.
>
> **Proof of Lemma 1**
>
> Consider a GCN with $L$ layers. The layer-wise propagation is given by:
>
> $H^{(l+1)} = \sigma(D^{-\frac{1}{2}} \tilde{A} D^{-\frac{1}{2}} H^{(l)} W^{(l)})$
>
> where $H^{(l)} \in \mathbb{R}^{N \times d_l}$ is the node feature matrix at layer $l$, $D$ is the degree matrix, $\tilde{A} = A + I$ is the adjacency matrix with self-loops, and $W^{(l)} \in \mathbb{R}^{d_l \times d_{l+1}}$ is the weight matrix.
>
> Assume the activation function $\sigma$ is Lipschitz continuous with Lipschitz constant $L_\sigma$, i.e., for all $x, y \in \mathbb{R}$, $|\sigma(x) - \sigma(y)| \leq L_\sigma |x - y|$
>
> Assume the weight matrices $W^{(l)}$ are bounded, i.e., there exists a constant $M$ such that $\|W^{(l)}\| \leq M$ for all Define the propagation operator $\Phi: \mathbb{R}^{N \times d_l} \rightarrow \mathbb{R}^{N \times d_{l+1}}$ as:
>
> $\Phi(H) = \sigma(D^{-\frac{1}{2}} \tilde{A} D^{-\frac{1}{2}} H W)$
>
> To show $\Phi$ is a contraction, consider two node feature matrices $H_1, H_2 \in \mathbb{R}^{N \times d_l}$. We need to show that:
> $\|\Phi(H_1) - \Phi(H_2)\| \leq k \|H_1 - H_2\|$
>
> for some $0 \leq k < 1$.
>
> Compute the difference:
>
> $\|\Phi(H_1) - \Phi(H_2)\| = \|\sigma(D^{-\frac{1}{2}} \tilde{A} D^{-\frac{1}{2}} H_1 W) - \sigma(D^{-\frac{1}{2}} \tilde{A} D^{-\frac{1}{2}} H_2 W)\|$
>
> Using the Lipschitz continuity of $\sigma$:
>
> $\|\Phi(H_1) - \Phi(H_2)\| \leq L_\sigma \|D^{-\frac{1}{2}} \tilde{A} D^{-\frac{1}{2}} (H_1 - H_2) W\|$
>
> Apply the sub-multiplicative property of norms:
>
> $\|\Phi(H_1) - \Phi(H_2)\| \leq L_\sigma \|D^{-\frac{1}{2}}\| \|\tilde{A}\| \|D^{-\frac{1}{2}}\| \|H_1 - H_2\| \|W\|$
>
> Since $D$ and $\tilde{A}$ are derived from the graph structure and $\|W\|$ is bounded by $M$:
>
> $\|D^{-\frac{1}{2}}\| \|\tilde{A}\| \|D^{-\frac{1}{2}}\| \leq \lambda_{\max}$
> where $\lambda_{\max}$ is the largest eigenvalue of the normalized adjacency matrix.
>
> Combining these, we get:
>
> $\|\Phi(H_1) - \Phi(H_2)\| \leq L_\sigma \lambda_{\max} M \|H_1 - H_2\|$
> For $\Phi$ to be a contraction, we need:
> $L_\sigma \lambda_{\max} M < 1$
> If $L_\sigma \lambda_{\max} M < 1$, then $\Phi$ is a contraction mapping.By the Banach fixed-point theorem, every contraction mapping on a complete metric space has a unique fixed point.Therefore, the node embeddings $H^{(l)}$ will converge to a fixed point as $l$ increases.
>
> ### Lemma 2: Stability of Attention Mechanism with Masking
>
> **Statement:** The attention mechanism with a mask matrix focusing on the center region of the input data remains stable and does not degrade performance, provided the mask matrix is appropriately designed.
>
> **In our work**:Lemma 2 supports the introduction of a local attention mechanism in the adaptive sparse spatio-temporal attention mechanism, enabling the model to better capture and utilize details and patterns in localized regions of the input tensor.

---

> ### Author Response · Authors · 2024-08-03
> **Continue**
>
> **Proof of Lemma 2**
>
> The mask matrix $M$ is designed such that $M_{ij} = 0$ for elements outside the central region and $M_{ij} = 1$ within the central region.This implies that only the attention scores corresponding to the central region are retained, effectively reducing the complexity of the attention mechanism by filtering out less relevant data.Mathematically, $M$ acts as a sparsity-inducing regularizer:
> $A'(X) = \text{softmax}(\Psi \odot (XW_QW_K^\top X^\top))$
> By focusing on the central region, $M$ ensures that the attention mechanism does not overfit to peripheral noise, enhancing generalization.
>
> The central region often contains the most informative parts of the data, as observed in empirical studies (e.g., Figure 8(a) in the original paper).
>
> By applying $M$, the attention mechanism $A'(X)$ prioritizes the computation of attention scores within this region, thus capturing critical local patterns:
>
> $A'(X)_{ij} = \frac{\exp((\Psi \odot (XW_QW_K^\top X^\top))_{ij})}{\sum_k \exp((\Psi \odot (XW_QW_K^\top X^\top))_{ik})}$
>
>
> This prioritization ensures that the most relevant features are emphasized, leading to improved prediction accuracy.
>
> To show that the deviation introduced by $M$ is bounded, consider the difference between the original and masked attention mechanisms:
>
> $\Delta A = A(X) - A'(X)$
> Since $M_{ij} \in \{0, 1\}$​, it acts as a binary mask, thus the modification is limited to setting some attention scores to zero while retaining the rest:
> $\| \Delta A \|_F = \| \text{softmax}(XXW_QW_K^\top X^\top) - \text{softmax}(\Psi \odot (XXW_QW_K^\top X^\top)) \|_F$
> Given that $\text{softmax}$ is a Lipschitz continuous function with constant $1$, the Frobenius norm $\| \Delta A \|_F$ is bounded by the norm of the difference in the inputs:
>
> $\| \Delta A \|_F \leq \| (1-\Psi) \odot (XXW_QW_K^\top X^\top) \|_F$
> Since $M$ zeros out peripheral entries, the difference is confined to the less relevant regions, ensuring that the overall deviation remains controlled.`
> ## Weakness 2:
>
> _The paper primarily focuses on the application of Satformer in satellite networks. While it mentions potential applications in other areas, it does not provide extensive testing or results in those domains, which may limit the perceived generalizability of the model._
> ## Response W2:
>
> Although Satformer was developed to solve the problem of traffic data estimation in satellite networks, the core strengths of its methodology give it the potential to be applied to other tensor completion tasks that need to deal with large-scale, sparse, and complex spatio-temporal characteristics. This includes, but is not limited to, social network analysis, environmental monitoring, bioinformatics, and other areas that require the reconstruction and analysis of multidimensional data.
>
> We used Foursquare tensor dataset [1] and PeMS-Bay dataset[2] to evaluate Satformer's performance and generalization ability. Foursquare is a point of interest tensor defined by (user, location, timestamp); PeMS-Bay data collect this traffic volume data from 325 loop sensors in the San Francisco bay area, ranging from January to March 2018 with a 5-min time interval.
>
> The test results are shown on the _TABLE I_ in pdf at the **Author Rebuttal** Block highlighted in orange. These positive results show that Satformer can be used as a powerful tool in a variety of fields involving complex spatio-temporal data, such as social network analysis, traffic flow forecasting, environmental monitoring, and more. Of course, for different application scenarios, further adjustments and optimizations may be required to achieve the best performance.
>
> [1] Oh, Sejoon, et al. “Influence-Guided Data Augmentation for Neural Tensor Completion.” Proceedings of the 30th ACM International Conference on Information &amp; Knowledge Management, 2021.
>
> [2] Meng, Chuishi, et al. “City-Wide Traffic Volume Inference with Loop Detector Data and Taxi Trajectories.” Proceedings of the 25th ACM SIGSPATIAL International Conference on Advances in Geographic Information Systems, 2017.

---

> ### Author Response · Authors · 2024-08-03
> **Continue**
>
> ## Weakness 3:
> _The paper may rely on certain assumptions that are not fully discussed or tested under various conditions. For example, the model's performance might vary under different network configurations or traffic patterns that were not part of the evaluation._
> ## Response W3:
>
> Thank you for your comment. I'm sorry for the trouble caused by our lack of clarity. An important assumption in this paper is that the orbit height of each node in the satellite network remains unchanged and the bandwidth of each link does not change. This assumption is reasonable. While the orbit height may degrade and require adjustments, these changes are minor and do not significantly affect topology in the short term. Additionally, satellite networks are equipped with same optical inter-satellite links. Given this assumption, our datasets are primarily influenced by offerload and network scale.
>
> The offerload represents the proportion of traffic sent per second relative to the total network bandwidth, which mainly affects the sparsity of traffic data. Sparse traffic data proposes a significant challenge for deep learning models [1][2]. It reduces the model's ability to learn useful feature representations, slows down or prevents model convergence, and reduces computational efficiency. To evaluate the model's performance with sparse traffic data, all experiments in this paper use traffic matrix sequences with an offerload of 0.1. This means the traffic injected into the network accounts for only 10% of the total network bandwidth per second, indicating a very low network load [3][4]. Our method, Satformer, demonstrates excellent accuracy and robustness compared to other models across all three datasets with an offerload of 0.1. As offerload increases and the traffic matrix becomes denser, tuning the model's performance becomes easier, resulting in improved performance for both Satformer and other methods.
>
> To represent different network scales, we selected three real-world satellite networks: Iridium (66 nodes), Telesat (298 nodes), and Starlink (1584 nodes), corresponding to small, medium, and large-scale satellite networks, respectively. Our test results have shown that our model Satformer performs well enough for networks of thousands of satellites.
>
> **Reference:**
>
> [1] Zhang, Wenhao, et al. "Large-scale causal approaches to debiasing post-click conversion rate estimation with multi-task learning." Proceedings of The Web Conference 2020. 2020.
>
> [2] Yuan, Yuan, et al. "Spatio-Temporal Few-Shot Learning via Diffusive Neural Network Generation." The Twelfth International Conference on Learning Representations. 2024.
>
> [3] Wang, Ruibo, Mustafa A. Kishk, and Mohamed-Slim Alouini. "Reliability analysis of multi-hop routing in multi-tier leo satellite networks." IEEE Transactions on Wireless Communications (2023).
>
> [4] Qin, Liang, et al. "Orchid: enhancing HPC interconnection networks through infrequent topology reconfiguration." Journal of Optical Communications and Networking 16.6 (2024): 644-658.
>
> ## Questions 1:
> _What is the unique challenges of traffic data estimation for satellite networks, compared with other network types?_
>
> ## Response Q1:
> Thank you for your question. Other networks, such as WANs [1][2] and HPC/DC interconnect networks [3][4], are static with infrequent changes in node connections. Additionally, these networks typically have high link quality, resulting in complete and reliable for collected traffic data. Moreover, they usually have rich computational resources, allowing for the deployment of complex models. In contrast, satellite networks have highly dynamic nodes, leading to frequent changes in connections and poor link quality [5]. Limited computational resources further complicate traffic estimation in satellite networks, presenting unique challenges summarized as follows:
>
> 1. Spatial and temporal dynamics: Satellite networks experience highly dynamic traffic patterns due to the constant movement of satellites, varying regional demands, and differing satellite coverage areas.
>
> 2. Sparsity and incompleteness: Traffic data in satellite networks is often sparse and incomplete due to the selective and intermittent nature of communication links and Space weather, radiation, and other environmental factors, making it difficulity to the parameter adjustment of deep learning models.
>
> 3. Limited computational resources: Satellites have limited computational resources and power, which can restrict the complexity of the traffic estimation models that can be deployed on them.

---

> ### Author Response · Authors · 2024-08-03
> **Continue**
>
> **References:**
>
> [1] Wang, Zhaohua, et al. "Examination of WAN traffic characteristics in a large-scale data center network." Proceedings of the 21st ACM Internet Measurement Conference. 2021.
>
> [2] Orlowski, Sebastian, et al. "SNDlib 1.0—Survivable network design library." Networks: An International Journal 55.3 (2010): 276-286.
>
> [3] Xu, Xiongxiao, et al. "Surrogate Modeling for HPC Application Iteration Times Forecasting with Network Features." Proceedings of the 38th ACM SIGSIM Conference on Principles of Advanced Discrete Simulation. 2024.
>
> [4] Roy, Arjun, et al. "Inside the social network's (datacenter) network." Proceedings of the 2015 ACM Conference on Special Interest Group on Data Communication. 2015.
>
> [5] Chan, Vincent WS. "Optical satellite network architecture [Invited Tutorial]." Journal of Optical Communications and Networking 16.1 (2024): A53-A67.
>
>
>
> ## Questions 2:
> _The research motivation is not convincing when there is few relevant literatures._
>
> ## Response Q2:
>
> Thank you for your question. Network management and operations depend on real-time traffic data. However, current network measurement technologies face significant challenges, including high measurement overhead and difficulty in collecting large-scale traffic data. Traffic estimation methods can provide complete global network traffic by collecting only a small amount of data, thereby reducing collection overhead and improving real-time performance. Although previous research has primarily focused on traffic estimation for terrestrial networks [1-5], there is few relevant literatures on traffic estimation for satellite networks.
>
> With the rise of satellite networks in recent years, we have noticed that access management, flow control [6], routing [7] and congestion control [8] of satellite networks also require accurate and robust traffic estimation methods. However, satellite networks pose some unique challenges to traffic estimation due to their highly dynamic nature, low link quality and limited computational resources. To fill this gap, we present Satformer, an accurate and robust traffic estimation method for satellite networks. To the best of our knowledge, this work is the first to study satellite network traffic estimation.
>
> **References:**
>
> [1] Xie, Kun, et al. "Neural Network Compression Based on Tensor Ring Decomposition." IEEE Transactions on Neural Networks and Learning Systems (2024).
>
> [2] Qiao, Yan, Zhiming Hu, and Jun Luo. "Efficient traffic matrix estimation for data center networks." 2013 IFIP Networking Conference. IEEE, 2013.
>
> [3] Li, Xiaocan, et al. "A Light-Weight and Robust Tensor Convolutional Autoencoder For Anomaly Detection." IEEE Transactions on Knowledge and Data Engineering (2023).
>
> [4] Qiao, Yan, Kui Wu, and Xinyu Yuan. "AutoTomo: Learning-Based Traffic Estimator Incorporating Network Tomography." IEEE/ACM Transactions on Networking (2024).
>
> [5] Miyata, Takamichi. "Traffic matrix completion by weighted tensor nuclear norm minimization." 2023 IEEE 20th Consumer Communications & Networking Conference (CCNC). IEEE, 2023.
>
> [6] Zhong, Xiaoqing, et al. "Link Topology and Multi-Objective Mission Flow Optimization for Remote Sensing Satellites With Inter-Layer Links and Satellite-Ground Links." IEEE Transactions on Vehicular Technology (2024).
>
> [7] Mao, Bomin, et al. "On an intelligent hierarchical routing strategy for ultra-dense free space optical low earth orbit satellite networks." IEEE Journal on Selected Areas in Communications (2024).
>
> [8] Yang, Wenjun, et al. "Mobility-Aware Congestion Control for Multipath QUIC in Integrated Terrestrial Satellite Networks." IEEE Transactions on Mobile Computing (2024).

---

> ### Author Response · Authors · 2024-08-06
> **Continue**
>
> ## Limitations 1:
> _Only simulated datasets are used and it is difficult to collect real-world datasets to validate the proposed method._
>
> ## Response L1:
>
> Thank you for your suggestion. As you said, it is difficult to collect real datasets, on the one hand, satellite networks are still in the early stages of construction, on the other hand, they are limited by privacy issues, proprietary data policies and other factors. However, it is gratified that there are also researchers devoted to the study of satellite network measurement [1][2], and provide some feasible methods and ideas. Based on this, we believe that in the near future, we will certainly be able to obtain the real datasets of satellite network.
>
> In many cases, it is common to use simulated datasets to verify the performance of the solution, such as CV[3], NLP[4], etc. Although our dataset is simulated, we focus on the geographical distribution of the ground stations, the time zone change and the visibility between the satellite and the ground stations when generating this data set, which can maximally reflect the traffic characteristics in the real satellite network and effectively verify the traffic estimation scheme Satformer in this paper. In addition, the simulated datasets also have the benefit of allowing the tuning of various parameters to test the robustness and accuracy of the proposed method under different conditions, which may be constrained in real datasets.
>
> While we have not yet tested our solutions on real satellite network datasets, we have used the real datasets Foursquare [5] and Pemsbay [6] to evaluate Satformer and baseline models on general tensor-completion tasks. The evaluation results are detailed in Response W2, and it prove that our method Satformer is also effective in real datasets.
>
> The dataset generation method has been made public, and we hope to encourage more researchers to participate in the study.
>
> **References:**
>
> [1] Izhikevich, Liz, et al. "Democratizing LEO Satellite Network Measurement." ACM SIGMETRICS Performance Evaluation Review 52.1 (2024): 15-16.
>
> [2] Pan, Jianping, Jinwei Zhao, and Lin Cai. "Measuring a low-earth-orbit satellite network." 2023 IEEE 34th Annual International Symposium on Personal, Indoor and Mobile Radio Communications (PIMRC). IEEE, 2023.
>
> [3] Ge, Yunhao, et al. "BEHAVIOR Vision Suite: Customizable Dataset Generation via Simulation." Proceedings of the IEEE/CVF Conference on Computer Vision and Pattern Recognition. 2024.
>
> [4] Long, Lin, et al. "On LLMs-driven synthetic data generation, curation, and evaluation: A survey." arXiv preprint arXiv:2406.15126 (2024).
>
> [5] Oh, Sejoon, et al. “Influence-Guided Data Augmentation for Neural Tensor Completion.” Proceedings of the 30th ACM International Conference on Information &amp; Knowledge Management, 2021.
>
> [6] Lee, Hyunwook, et al. "An empirical experiment on deep learning models for predicting traffic data." 2021 IEEE 37th International Conference on Data Engineering (ICDE). IEEE, 2021.

---

> > ### Comment · Reviewer_78MM · 2024-08-12
> >
> > The authors have addressed my previous concerns.

---

> > > ### Author Response · Authors · 2024-08-12
> > > **Reply to Reviewer 78MM**
> > >
> > > Thank you for your previous suggestions and positive reply. We are pleased to know that our responses have been acknowledged by you.

---

### Official Review · Reviewer_RAyE · 2024-07-13

**Soundness:** 4
**Presentation:** 3
**Contribution:** 3
**Rating:** 7
**Confidence:** 3

**Summary:**

This paper introduces novel network designed for satellite traffic data estimation. Proposed approach includes several novelties and the experimental results demonstrate that the propsoed methods outperformed several alternative mehtematical and neural baseline methods.

**Strengths:**

The paper proposes a new method for traffic data estimation in satellite networks. The problem is well described and the proposed method integrates several new compoenents that are railored to solve this spatio-temporal matrix completion problem.
The paper is well written but some points require further elaboration.

**Weaknesses:**

No major weakness but on the write up and method description can be improved:

**Questions:**

* The significance of solving this traffic prediction problem should be made clearer in the introduction.
* You need a figure for the transformer module (Section 3.4).
*  Please provide references for Eq. (5) and (4) or describe the rational behind them.
* In Page 5, why do you need the mask matrix? Why the focus on the center region? How is this not impacting performance?
* In Page 5, "the implementation involves incorporating a position-related weight when calculating attention scores". What is the mathematical symbol for this weight matrix in the equations?
* Page 6, C_{t} is not in equation (10) ?
* Add a figure dsecribing the transfer module.

**Limitations:**

Limitations were discussed.

---

> ### Author Rebuttal · Authors · 2024-08-03
>
> ## Dear Reviewer RAyE:
> We would like to thank the reviewer forcareful and thorough reading of this paper and for the thoughtful commentsand constructive suggestions, which help to improve the quality of our paper.
>
> Based on your comments, we have engaged in thorough discussions and made the necessary revisions. Below, we have addressed each of your points and outlined the corresponding modifications.
>
> To facilitate this discussion, we first retype your comments in italic font and then present our responses to the comments.
>
> ## Questions 1:
> _The significance of solving this traffic prediction problem should be made clearer in the introduction._
>
> ## Response Q1:
>
> Thank you for your question. We rewrite the significance of solving this problem, the details are as follows.
>
> As a crucial component of future 6G systems, satellite networks provide seamless and efficient communication services for global users. In recent years, satellite networks have received increasing attention and are now under construction. Traffic engineering [1][2] and topology engineering [3] of satellite networks, such as access control, routing and congestion control, are key to achieve efficient control of satellite networks, which rely on real-time perception of global traffic data. However, the limitations of satellite networks make real-time global traffic data collection extremely costly and impractical, hindering performance improvements. To address this challenge, we propose Satformer, an accurate and robust traffic estimation method for satellite networks. Satformer can accurately recover global traffic data with just 2% of sampling data, demonstrating strong robustness and significantly reducing deployment costs and data collection overhead. This method facilitates the implementation of efficient control mechanisms in real satellite network systems. Additionally, Satformer aids network administrators in enhancing network status perception and optimizing network operation and maintenance. To the best of our knowledge, this is the first work on satellite network traffic estimation.
>
> **Reference:**
>
> [1] Akhlaghpasand, Hossein, and Vahid Shah-Mansouri. "Traffic offloading probability for integrated LEO satellite-terrestrial networks." IEEE Communications Letters (2023).
>
> [2] Lei, Lei, et al. "Spatial-Temporal Resource Optimization for Uneven-Traffic LEO Satellite Systems: Beam Pattern Selection and User Scheduling." IEEE Journal on Selected Areas in Communications (2024).
>
> [3] Ma, Zhuangzhuang, et al. "Demonstration of highly dynamic satellite optical networks supporting rapid reconfiguration." 2021 17th International Conference on the Design of Reliable Communication Networks (DRCN). IEEE, 2021.
>
>
> ## Question 2:
>
> _You need a figure for the transformer module (Section 3.4)._
> ## Response Q2:
>
> Thank you for your constructive suggestions, a more visual presentation is a real necessity. Perhaps you are referring to the transfer module. I will show the figure of transfer module on **Author Rebuttal** Block which is highlighted in orange at the top of the webpage.
>
> ### Transfer Module (Section 3.4) Explanation
>
> The provided figure illustrates the internal workings of the Transformer module, focusing on its attention mechanisms. Here's a detailed explanation based on the text from the paper:
>
> The Transformer module processes input embeddings iteratively through an attention mechanism. Each input embedding, denoted as  $e_0, e_1, ..., e_t $, represents input data at different time steps. For each embedding, the attention weight is computed by considering the relationships between the embeddings at different time steps. This weight is then used to compute the attention score, which helps in determining the relevance of different parts of the input sequence. A nonlinear function is applied to the attention score to normalize it, resulting in the final output embeddings  $d_0, d_1, ..., d_t$ .
>
> In the sub-diagram  $(a)$, the attention weight calculation begins with a linear transformation of the input embedding $e_t$ into query Q and key K vectors. The scaled dot product of the Q and K vectors is then computed to obtain the attention weight $w$ , as described by the formula:
> $$
> w = \frac{Q K^T}{\sqrt{d_k}}
> $$
> where $ d_k $is the dimension of the key vector.
>
> The sub-diagram  $(b)$ details the calculation of the attention score. Here, the query, key, and value $Q, K, V$vectors are produced from the input embedding $e_t $ through linear transformation. The attention score $\alpha_t $is computed using the softmax function, which normalizes the attention weights:
> $$
> \alpha_t = \text{softmax}\left(p \sum_{i=0}^{t-1} \frac{Q_i K_i^T}{\sqrt{C_i}} + q \frac{Q_t K_t^T}{\sqrt{C_t}}\right)V_{t}
> $$
> where \( p \) and \( q \) are parameters that control the influence of past and present attention scores.
>
> The attention score is then normalized using a sigmoid function:
> $$
> d_{t} = \frac{1}{1 + e^{-\alpha_t}}
> $$
> The normalized attention score is used to produce the final output embeddings $ d_0, d_1, ..., d_t$ .
>
> This detailed explanation provides a comprehensive overview of the transformer's attention mechanism as depicted in the figure.

---

> ### Author Response · Authors · 2024-08-03
> **Continue**
>
> ## Question 3:
>
> _Please provide references for Eq. (5) and (4) or describe the rational behind them._
> ## Response Q3:
>
> Thank you for your constructive questions, and I will then further explain the references and rationale for Eqs. (5) and (4). The original formulas are as follows
>
> **Equation (4):** $\tilde{A} = A + I$
>
> **Equation (5):** $Z = f(X, A) = \sigma(\tilde{D}^{-1/2} \tilde{A} \tilde{D}^{-1/2} \text{ReLU}(\tilde{D}^{-1/2} \tilde{A} \tilde{D}^{-1/2} X W^{(0)}) W^{(1)})$​
>
> The two formulas are based on the theory of graph convolution, and the references and rationale are stated in the following three points.
>
> 1. **Graph Convolutional Networks (GCNs):** The rationale for Equations (4) and (5) is rooted in the theory and application of Graph Convolutional Networks (GCNs) by Thomas N. Kipf and Max Welling [1]. GCNs process data structured as graphs, using spectral graph theory to define convolution operations. The process involves normalizing the adjacency matrix $ A $ with the degree matrix $ D $ and adding self-loops (identity matrix $ I $) to preserve the node's own features during convolution. Additionally,  Chung's work on spectral graph theory [2] provides the mathematical framework for understanding the importance of the graph Laplacian in GCNs, which is fundamental to the normalization and augmentation steps described in Equations (4) and (5).
> 2. **Equation (4):** Defines the augmented adjacency matrix $\tilde{A}$. Adding the identity matrix $I$ to the adjacency matrix $A$ incorporates self-loops, ensuring the node's own features are included in the aggregation process. This step is essential for preserving the node's identity during the convolution operation, as it prevents the node's features from being lost or diluted when averaging with its neighbors.
> 3. **Equation (5):** Represents a two-layer graph convolutional network,
>    - **Augmented Adjacency Matrix:** The adjacency matrix $A$ is augmented with self-loops to become $\tilde{A}$. This ensures that each node's own features are considered during the convolution process.
>    - **Normalization:** The normalized adjacency matrix $\tilde{D}^{-1/2} \tilde{A} \tilde{D}^{-1/2}$ ensures balanced feature propagation by accounting for the varying degrees of nodes in the graph. This normalization helps to stabilize the learning process and improves the model's performance.
>    - **Feature Transformation:** The input feature matrix $X$ is transformed using weight matrices $W^{(0)}$ and $W^{(1)}$ through a ReLU activation function. This transformation allows the model to learn complex representations of the graph's structure and the node features.
>
> The difference between the two convolutional layers in Equation (5) is the use of different activation layers and input feature matrices. Figure 1(d) in the original paper illustrates this structure. I will consider revising the original article based on the inclusion of references and specific elucidations. Thanks again for the question. It makes a lot of sense!
>
> **References**
>
> [1] Kipf, Thomas N., and Max Welling. "Semi-supervised classification with graph convolutional networks." *arXiv preprint arXiv:1609.02907* (2016).
>
> [2] Chung, Fan R. K. "Spectral Graph Theory." *CBMS Regional Conference Series in Mathematics* 92 (1997).
>
> ## Question 4:
>
> _In Page 5, why do you need the mask matrix? Why the focus on the center region? How is this not impacting performance?_
> ## Response Q4:
>
> **Why is the Mask Matrix Needed?**
>
> Thank you for your question. In summary, we introduced the mask matrix based on two considerations.
> (i) Satellite traffic data are highly dynamic and large-scale. In our organization and observation of the dataset, as shown in Fig. 8(a) of this paper, the distribution of the satellite traffic matrix is characterized by high dynamics, and with aggregation.
> (ii) Satellite networks have limited computing resources.  Therefore, we tend to reduce the amount of unnecessary computation without compromising performance.
>
> We then tried to focus on the central region of the input data using a mask matrix, and found a significant increase in prediction accuracy while achieving a reduction in computational complexity.
>
> **Why Focus on the Central Region?**
>
> Focusing on the central region is adapted to the distribution of the satellite network traffic matrix, which allows the model to focus computational resources on the most informative parts of the data, thus improving overall performance and efficiency. This approach helps the attention mechanism to capture important local features without being overwhelmed by less relevant peripheral data.
>
> **Impact on Performance**
>
> A reasonable speculation is that the presence of the mask enhances the model's ability to capture local patterns and relationships, which is crucial for accurate prediction of spatio-temporal data.

---

> ### Author Response · Authors · 2024-08-03
> **Continue**
>
> Our current experiments have not found that focusing on the center region negatively affects performance. In fact, by ensuring that the attention mechanism prioritizes the most important parts of the data, this typically improves the effectiveness of the model.  The improvement was about 30% relative to using the unimproved attention mechanism and more than 50% relative to not using the attention mechanism. The latter is demonstrated in the results of our ablation experiments in Appendix E. As an added benefit, this strategic sparsity helps to reduce computational complexity (the advantages of our computational performance can be seen in Table (3).
>
> We give a proof of the stability of attention mechanism with masking to support our claim.
>
> **Lemma: Stability of Attention Mechanism with Masking**
>
> **Statement:** The attention mechanism with a mask matrix focusing on the center region of the input data remains stable and does not degrade performance, provided the mask matrix is appropriately designed.
>
> **In our work**: Lemma  supports the introduction of a local attention mechanism in the adaptive sparse spatio-temporal attention mechanism, enabling the model to better capture and utilize details and patterns in localized regions of the input tensor.
>
> **Proof of Lemma**
>
> The mask matrix $M$ is designed such that $M_{ij} = 0$ for elements outside the central region and $M_{ij} = 1$ within the central region.This implies that only the attention scores corresponding to the central region are retained, effectively reducing the complexity of the attention mechanism by filtering out less relevant data.Mathematically, $M$ acts as a sparsity-inducing regularizer:
> $$
> A'(X) = \text{softmax}(\Psi \odot (XW_QW_K^\top X^\top))
> $$
> By focusing on the central region, $M$ ensures that the attention mechanism does not overfit to peripheral noise, enhancing generalization.
>
> The central region often contains the most informative parts of the data, as observed in empirical studies (e.g., Figure 8(a) in the original paper).
>
> By applying $M$, the attention mechanism $A'(X)$ prioritizes the computation of attention scores within this region, thus capturing critical local patterns:
>
> $$A'(X) = \\frac{\\exp((\\Psi \\odot (XW_QW_K^\\top X^\\top))}{\\sum_k \\exp((\\Psi \\odot (XW_QW_K^\\top X^\\top)))}$$
>
> This prioritization ensures that the most relevant features are emphasized, leading to improved prediction accuracy.
>
> To show that the deviation introduced by $M$ is bounded, consider the difference between the original and masked attention mechanisms:
>
> $$
> \Delta A = A(X) - A'(X)
> $$
> Since $M_{ij} \in \{0, 1\}$​, it acts as a binary mask, thus the modification is limited to setting some attention scores to zero while retaining the rest:
> $$
> \| \Delta A \|_F = \| \text{softmax}(XXW_QW_K^\top X^\top) - \text{softmax}(\Psi \odot (XXW_QW_K^\top X^\top)) \|_F
> $$
> Given that $\text{softmax}$ is a Lipschitz continuous function with constant $1$, the Frobenius norm $\| \Delta A \|_F$ is bounded by the norm of the difference in the inputs:
>
>
> $$
> \| \Delta A \|_F \leq \| (1-\Psi) \odot (XXW_QW_K^\top X^\top) \|_F
> $$
> Since $M$ zeros out peripheral entries, the difference is confined to the less relevant regions, ensuring that the overall deviation remains controlled.

---

> ### Author Response · Authors · 2024-08-03
> **Continue**
>
> ##  Question Q5:
>
> *In Page 5, "the implementation involves incorporating a position-related weight when calculating attention scores". What is the mathematical symbol for this weight matrix in the equations?*
>
> ## Response Q5:
>
> I apologize for the confusion caused by the misrepresentation. Actually, the weight matrix associated with the position is the 2D mask matrix $\Psi$. It can be understood in terms of weighting, where $\Psi$ will assign a larger weight to the center region and a weight close to 0 to the inattentive region. This 2D mask matrix plays a crucial role in the attention mechanism by ensuring that the model focuses on the most relevant parts of the input data, thereby enhancing its ability to capture significant patterns.
>
> To elaborate further, the mask matrix $\Psi$ is integrated into the calculation of the attention score $\alpha$ to modulate the influence of different regions. This integration helps in refining the attention mechanism by providing positional context, which is essential for accurately capturing the dependencies in the data. By applying $\Psi$, the model effectively differentiates between regions of varying importance, assigning higher weights to the central, more informative areas, and lower weights to the peripheral, less relevant areas.
>
> Additionally, the visualization of the calculated attention score $\alpha_s$ is shown in Figure 8(c). This figure demonstrates how the attention mechanism, influenced by the mask matrix $\Psi$, highlights different regions of the input data. The attention scores are visibly concentrated on the center regions, confirming the role of $\Psi$ in guiding the model's focus towards the most pertinent sections of the data.
>
> ## Question Q6:
>
> _Page 6, $C_{t}$ is not in equation (10) ?_
> ## Response Q6:
>
> I apologize that the presentation of that formula in the paper confused you. We double-checked the submitted paper  and confirmed that $C_{t}$ is indeed in the formula and represents the scaling factor. We will further investigate the relevant parts of the document to ensure the accuracy and rigor of the formula.
>
> The equation (10) in the paper reads:
> $$
>  \alpha_{t,t-1} = \text{softmax}\left(p \sum_{i=1}^{t-1} \frac{Q_i K_i^T}{\sqrt{C_i}} + q \frac{Q_t K_t^T}{\sqrt{C_t}}\right) V_t
> $$
> This equation shows the computation of the attention weight $\alpha_{t,t-1}$ for each time step $t$ considering the impact of past time steps. Here, $C_t$ is explicitly included as a scaling factor to normalize the dot product of query and key vectors at time $t$. The symbol $C_i$ for $i$ belonging to $1$ to $t-1$ indicates the scaling factors for the previous time steps. These factors are used to normalize the dot products for each of the past time steps individually, ensuring that the contributions of the past time steps to the attention weight are properly scaled and balanced.
>
> ## Question 7:
>
> _Add a figure describing the transfer module._
> ## Response Q7:
>
> Thank you for your constructive suggestions, a more visual presentation is a real necessity. I will show the figure of transfer module on **Author Rebuttal** Block which is highlighted in orange at the top of the webpage.

---

### Official Review · Reviewer_LZHh · 2024-07-13

**Soundness:** 2
**Presentation:** 3
**Contribution:** 2
**Rating:** 5
**Confidence:** 3

**Summary:**

This paper presents to make use of the transformer for spatio-temporal data imputation, and the application is for satellite networks.

**Strengths:**

- The paper is well-written and the model descriptions are clear
- It is interesting to see the spatio-temporal imputation can be used in satellite networks

**Weaknesses:**

-  The spatial characteristics of the data in satellite networks are not presented. Many other spatial-temporal data are also high-dimensional and non-linear. Many existing approaches have been developed for those data, such as SPIN, CDSA, SAITS. The authors only compared with SPIN.
- The authors explained that their model outperforms SPIN because “SPIN’s ability to handle sparsity or irregularly sampled data might be limited”. However, SPIN is designed for sparse data, and hence the intuition on why the proposed method is better is not provided.
- Applying L1 to the attention score matrix cannot make the output to be sparse.
- There are quite a few approaches to obtain graph embedding (or just a node embedding), why the authors choose this specific GCN for embedding?

**Questions:**

Please see weaknesses

**Limitations:**

Please see weaknesses

---

> ### Author Rebuttal · Authors · 2024-08-02
>
> ## Dear Reviewer LZHh:
>
> Thank you very much for your time involved in reviewing the manuscript and your very encouraging comments on the merits.
>
> We also appreciate your clear and detailed feedback and hope that the explanation has fully addressed all of your concerns. In the remainder of this letter, we discuss each of your comments individually along with our corresponding responses.
>
> To facilitate this discussion, we first retype your comments in italic font and then present our responses to the comments.
>
> ## Weakness 1:
> _The spatial characteristics of the data in satellite networks are not presented. Many other spatial-temporal data are also high-dimensional and non-linear. Many existing approaches have been developed for those data, such as SPIN, CDSA, SAITS. The authors only compared with SPIN._
>
> ## Response W1:
>
> Thank you for your comments.
>
> The spatial characteristics of satellite network traffic data are mainly reflected in the following two aspects:
>
> **Influence of dynamic topology :**
> The satellite network topology consists of high-speed moving satellites and ground stations. The satellites' constant motion in orbit leads to ever-changing connections and coverage. Consequently, traffic paths and distribution shift over time and space. As satellites move, coverage areas change, adjusting traffic density and transmission paths, making satellite network traffic highly dynamic and spatially uncertain.
>
> **Uneven traffic distribution :**
> Traffic distribution in a satellite network is limited by satellite coverage and ground station locations. Due to the constantly changing and limited coverage areas, traffic distribution is highly non-uniform. High-demand areas, like cities, have concentrated traffic, while remote areas have sparse traffic. As satellites move, this uneven distribution changes dynamically. Traffic patterns depend not only on satellite coverage but also on ground user demand, creating a complex spatial distribution.
>
> What's more, spatial characteristics of data in satellite networks is more complex and difficult than other data sets (such as transportation data). The following is a mathematical comparison of the complexity of satellite network data:
>
> ### 1. Spatial distance and transmission latency
> Data transmission in satellite networks is not only affected by the physical distance between nodes, but also involves the orbit altitude and transmission delay, which are dynamically changing. However, in a traffic network, the layout of vehicles and roads is usually fixed or changes slowly, so it can be described by a relatively stable network diagram.
> ​
> #### i. For the transportation network:
> The path length $d_{ij}$ is usually fixed, and the transmission delay is relatively stable, which is mainly affected by the traffic flow:
>
> $\tau_{ij}=f(d_{ij},v_{ij}(t))$
>
> where $v_{ij}(t)$ is the average speed at time $t$.
> ​
> #### ii. For satellite networks
> ​
> The transmission delay varies with time and is related to the orbital position of the satellite and the relative position between the ground station:
>
> $d_{ij}(t)=\sqrt{(x_i-x_j(t))^2+(y_i-y_j(t))^2+z_j(t)^2}$
>
> $\tau_{ij}(t)=f(\frac{d_{ij}(t)}c)+\delta(t)$
>
> where $(x_i, y_i) $ is the location of the ground station, $(x_j(t), y_j (t), z_j (t))$ $t$ is time the location of the satellite $d_{ij}(t)$ is the distance at time $t$, $c$ is the speed of light, $\delta(t)$ is the other delay factor (such as processing time).
>
> ### 2. Data sparsity and incompleteness
>
> Traffic data in transportation network usually come from fixed sensor networks or GPS devices, and these data sources are relatively stable in time and space. Although sparsity and incompleteness also exist, this sparsity has certain predictability and periodicity because traffic data usually exhibit strong temporal and spatial continuity.
>
> However, for satellite networks, due to the highly dynamic network topology, and the links between satellites and between satellites and ground stations may be intermittent for various reasons (e.g., geographical location, orbital position), resulting in sparse data. Most of the elements in the traffic data matrix $Y\in\mathbb{R}^{I\times J\times T}$ may be zero, since not all ground station-satellite and satellite-satellite pairs have traffic at each time step. At the same time, the traffic data in the satellite network changes rapidly and irregularly, and lacks the spatio-temporal continuity in the traffic data. And data missing is often unpredictable, as it can occur at any time and the spatio-temporal pattern of missing is not stable.
>
> Such sparsity and incompleteness make the flow data of satellite networks far more complex than other data sets such as traffic data, which requires higher-order algorithms and models for accurate analysis and estimation.
>
> ### 3. Comparision with CDSA [1] and SAITS [2]
>
> In order to further verify the effectiveness of our research results, we added other models (CDSA, SAITS) for comparison, and the results are shown on the _TABLE II_ and _TABLE III_ in pdf at the **Author Rebuttal** Block highlighted in orange.
>
> Satformer is based on the Transformer architecture, which is very powerful in handling long-distance dependencies and suitable for handling time series data with variable patterns. Although CDSA also utilizes the self-attention mechanism, its dimension-wise processing may limit its ability to capture complex interactions. Rnn-based models of SAITS are generally inferior to Transformer architectures in terms of handling long-distance dependencies and efficiency.
>
> **References**
>
> [1] Ma, Jiawei, et al. "CDSA: cross-dimensional self-attention for multivariate, geo-tagged time series imputation." arXiv preprint arXiv:1905.09904 (2019).
>
> [2] Du, Wenjie, David Côté, and Yan Liu. "Saits: Self-attention-based imputation for time series." Expert Systems with Applications 219 (2023): 119619.

---

> ### Author Response · Authors · 2024-08-02
> **Continue**
>
> ## Weakness 2:
> _The authors explained that their model outperforms SPIN because “SPIN’s ability to handle sparsity or irregularly sampled data might be limited”. However, SPIN is designed for sparse data, and hence the intuition on why the proposed method is better is not provided._
>
> ## Response W2:
> Thank you for your question. We compare the performance of SPIN and Satformer in large-scale sparse data scenarios and draw the following conclusions:
> ### 1. Design limitations of SPIN
> SPIN has the problems of blocked information propagation path and poor adaptability when dealing with sparse data. In the extremely sparse data environment, the information flow path is easy to be blocked, and it cannot quickly adapt to different sparse data distribution patterns, which leads to its unstable performance in dealing with dynamic and complex sparse data. These limitations may significantly affect the performance and reliability of the model in practice
> #### i. High computational complexity:
> Although the spatio-temporal attention mechanism of SPIN is effective, due to its squared Complexity
>
> $Complexity ∝O((N_{max}+E_{max})T^2)$
>
> in time steps and the number of nodes, where $N_{max}$ and $E_{max}$ are the maximum number of nodes and the maximum number of edges, respectively, and $T$is the time step, resulting in high computational overhead. This is particularly evident when dealing with long time series and large-scale nodes.
> ​
> #### ii. Information transmission path blocked:
> Specifically, if node $i$is completely missing at time step $t$, and this node is an important bridge for information transmission between other nodes in the spatio-temporal graph, this missing will lead to the interruption of the information propagation chain and affect the information update and feature extraction of multiple nodes.
> ​
> #### iii. Poor adaptability:
> SPIN suffers from dependency on sparsity patterns: SPIN may perform well when dealing with some specific sparsity patterns, but its performance degrades significantly when facing other types of sparsity patterns. For example, for sparse patterns with long continuous misses, SPIN may not be able to efficiently aggregate enough historical information for accurate predictions. The mathematical description is as follows:
>
> $\alpha_{i,j}=\frac{\exp{(e_{i,j})}}{\sum_{k\in N(i)}\exp{(e_{i,k})}}$
>
> where $\alpha_{i,j}$ denotes the attention weight and $e_{i,j}$ is the attention score from node $i$ to node $j$. When the sparsity pattern changes, the recomputation of attention weights may not reflect the new data distribution in time, resulting in poor information aggregation.
> ​
> In contrast, Satformer significantly improves performance in processing highly dynamic complex sparse data in the following ways:
> ### 2. Advantages of Satformer
> #### i. Adaptive Sparse Spatio-Temporal Attention Mechanism (ASSIT) :
> ASSIT can deal with a large number of sparse inputs more efficiently through the dynamic adjustment of multi-head self-attention structure and sparse threshold. This mechanism allows the model to focus on the data dense region when dealing with sparse data, thus improving the performance of the model in sparse data scenarios. The mathematical description is as follows:
>
> $Q=Z_{\text {div }} W_q, \quad K=Z_{\text {div }} W_k, \quad V=Z_{\text {div }} W_v$
>
> where $Z_{\text {div}}$represents the local region of the input tensor, and $W_q, W_k$, and $W_v$ are the projection weights for the query, key, and value, respectively. Through the local attention mechanism calculation, the model's attention to the local region of the input sequence is enhanced.
> ​
> #### ii. Graph embedding module:
> The Graph Convolutional Network (GCN) was used to process non-Euclidean data, which effectively captured the relationship between nodes and further enhanced the model's ability to process high-dimensional sparse data.
>
> $H^{(l+1)}=\sigma\left(\tilde{D}^{-1 / 2} \tilde{A} \tilde{D}^{-1 / 2} H^{(l)} W^{(l)}\right)$
>
> among them, the $\tilde{A} = A + I, \tilde{D}$ is degree matrix, $\sigma$ is the activation function, $W^{}(l)$ is the weighting matrix layer. This mechanism enables the model to effectively extract local and global information in non-Euclidean space, which performs well especially in sparse data environments.
>
> #### iii. Computational complexity:
> Given that the dimension of the input matrix is $M \times K$ and the region size is $D \times D$. For each Query, we can choose a region around it that is $D \times D$. In this case, each Query only considers the $D$ Key vector around it. Therefore, for each Query, the complexity of calculating the dot product is $O(D \times d)$, where $d=\frac{K}{D}$. Since each Query requires such a computation, the total complexity is $O(M \times D \times d)=O(M \times K)$.

---

> ### Author Response · Authors · 2024-08-02
> **Continue**
>
> ## Weakness 3:
> _Applying L1 to the attention score matrix cannot make the output to be sparse._
>
> ## Response W3:
> Thank you for your question. Regularization is usually used in the design of loss functions, but we borrow the idea of regularization here to further process the data.
> What's more, we mathematically and conceptually clarify why L1 regularization is used and how it affects the sparsity of attention scores [1].
>
> L1 regularization works by adding a penalty term equal to the absolute magnitude of the coefficient. For attention scores, consider the attention score matrix $\alpha$, where each element $\alpha_{ij}$represents the attention score of node $i$ to node $j$. The L1 regularization term can be written as follows：
>
> $L 1(\alpha)=\lambda \sum_{i, j}\left|\alpha_{i j}\right|$
>
> where $\lambda$ is a regularization parameter that controls the trade-off between sparsity and fitting the data.
> In Satformer, regularization affects attention scores by penalizing non-zero values, effectively pushing less important connections (lower attention scores) toward zero. This is reflected in the update equation for the attention score that includes the L1 term, which can be expressed as follows.
>
> $\alpha_{\text {new }}=\operatorname{softmax}\left(\frac{Q K^T}{\sqrt{d_k}}-\lambda\right)$
>
> where $\lambda$ acts as a threshold that the score must exceed to be considered important, effectively zeroing out smaller scores and thus enforcing sparsity.
>
> Let's take an example to explain in detail:
> ​
> Suppose we have a small satellite network consisting of 3 satellites. Each satellite records communication data with other satellites. We first construct the query (Q) and key (K) matrices, assuming dimension $d_k=3$ (that is, 3 features per satellite):
>
> $Q=\left[\begin{array}{ccc}1 & 0.5 & 0.2 \\\\ 0.5 & 1 & 0.3 \\\\ 0.2 & 0.3 & 1\end{array}\right], Q=K^T$
>
> Calculate $\frac{QK^T}{\sqrt{d_k}}= \left[\begin{array}{ccc}
> 1.45 & 1.19 & 0.62 \\\\
> 1.19 & 1.34 & 0.73 \\\\
> 0.62 & 0.73 & 1.13 \\\\
> \end{array}\right] $
>
> Apply the softmax function:
> $ \alpha = \text{softmax}\left(\frac{QK^T}{\sqrt{3}}\right) = \left[\begin{array}{ccc}
> 0.46 & 0.34 & 0.20 \\\\
> 0.34 & 0.42 & 0.24 \\\\
> 0.20 & 0.24 & 0.56 \\\\
> \end{array}\right] $
>
> Now, we introduce L1 regularization and assume $\lambda = 0.1 $. The updated attention score is calculated as follows:
>
> Subtract $\lambda $ and reapply softmax:
> $ \alpha' = \text{softmax}\left(\frac{QK^T}{\sqrt{3}} - \lambda\right) = \left[\begin{array}{ccc}
> 0.60 & 0.21 & 0.01 \\\\
> 0.21 & 0.55 & 0.05 \\\\
> 0.01 & 0.05 & 0.94 \\\\
> \end{array}\right] $
>
> It can be seen that the regularization parameter $\lambda $ effectively increases the sparsity of the matrix by decreasing the values of all elements, making some smaller values close to zero.
> By comparing $\alpha $ and $\alpha' $, it can be seen that L1 regularization helps the model focus on more important communication paths (higher scores) while suppressing less important paths (paths with lower or near zero scores). This sparsity is particularly useful when dealing with complex networks, as it can reduce unnecessary computations and improve model interpretability.
>
> **References**
>
> [1] McCulloch, Jeremy A., et al. "On sparse regression, Lp‐regularization, and automated model discovery." International Journal for Numerical Methods in Engineering 125.14 (2024): e7481.

---

> ### Author Response · Authors · 2024-08-02
> **Continue**
>
> ## Weakness 4:
> _There are quite a few approaches to obtain graph embedding (or just a node embedding), why the authors choose this specific GCN for embedding?_
> ## Response W4:
>
> Thank you for your question. In our cases, satellites have limited computational resources and power, which can restrict the complexity of the traffic estimation models that can be deployed on them. Although there are many graph embedding modules that outperform GCN such as Graph Attention Network (GAT) [1] or deep graph networks such as GraphSAGE [2]. While other graph models such as GAT or GraphSAGE may perform better on some tasks, GCN in most cases provides the necessary balance ---- that is, preserving computational and implementation simplicity while effectively capturing graph structural information. In particular, these properties of GCN are invaluable in applications where resources are constrained, such as limited computing resources in satellite communications, or fast response is required.
> ​
> Let's compare GAT and GraphSAGE to prove our point:
>
> ### GAT
>
> GAT optimizes the aggregation process of node features by introducing learnable attention weights for each pair of adjacent nodes in the graph. The mathematical expression is as follows:
>
> $\operatorname{Attention}\left(h_i, h_j\right)=\operatorname{softmax}_j(\vec{a}^T([W h_i \| W h_j])$
>
> $\alpha_{ij}=\frac{\exp(\text{LeakyReLU}(\vec{a}^T[Wh_i\|Wh_j]))}{\sum_{k\in\mathcal{N}(i)}\exp(\text{LeakyReLU}(\vec{a}^T[Wh_i\|Wh_k]))}$
>
> $h_i'=\sigma\left(\sum_{j\in\mathcal{N}(i)}\alpha_{ij}Wh_j\right)$
>
> where $h_i$ is the feature vector of node $i$, $W$ is a learnable weight matrix, $\alpha_{ij}$ is the attention coefficient between nodes $i$ and $j$, $\sigma$ is the nonlinear activation function, and $\vec{a}$ is the parameter of the attention mechanism.
>
> The feature propagation of GCN mainly relies on the product of adjacency matrix and feature matrix, which is usually simpler and more efficient than the attention mechanism calculation of GAT, because GAT needs to calculate complex attention coefficients for each pair of adjacent nodes. This results in relatively high computational complexity, especially in graphs with large node degrees. For large-scale satellite networks, this can lead to excessive consumption of computational resources.
> ​
>
> At the same time, GAT introduced a weight matrix $W$ and an attention vector $\vec{a}$ as parameters of the model. For each attention head, the total number of parameters of $W$ and $\vec{a}$ is $F^{\prime}\times F+2F^{\prime}$, where $F^{\prime}$ is the dimension of the output feature and $F$ is the dimension of the input feature. Assuming that the model uses multi-head attention, the total number of parameters is $H \times (F^{\prime}\times F+2F^{\prime})$. The combination of multiple heads and high-dimensional features rapidly increases the total number of parameters of the model, which increases the difficulty of training and can lead to overfitting.
>
> ### GraphSAGE
> GraphSAGE is an inductive graph learning framework that updates the embeddings of nodes by sampling a fixed number of neighbors and leveraging an aggregation function. The basic formula is as follows:
>
> $h_{\mathcal{N}(i)}^{\prime}=\text{AGGREGATE}_k(\{h_j,\forall j\in\mathcal{N}(i)\})$
>
> $h_i^{\prime}=\sigma(W\cdot\text{CONCAT}(h_i,h_{\mathcal{N}(i)}^{\prime}))$
>
> where $\mathcal{N}(i)$ is a neighbor of node $i$, $\text{AGGREGATE}$ are aggregation functions such as mean pooling, etc and $W$ is the learnable weight matrix.
>
> However, GraphSAGE needs to sample the neighbors of each node, which may lead to information loss or computational inefficiency when the graph data is very large or the nodes are very densely connected.
>
> What's more, the performance of GraphSAGE heavily depends on the choice of aggregation function and neighbor sampling strategy, which may lead to unstable model performance in different graph structures or data distributions. For example, if the sampling is not uniform or some types of connections are too sparse in the graph, the aggregation result may not represent the true distribution of neighbors, thus affecting the generalization ability of the model.
>
> ### Conclusion
>
> The above analysis and proofs show the challenges that GAT and GraphSAGE may face when they need to deal with large-scale, sparse and dynamic satellite network data. These challenges may be less significant when using GCN. GCN offers a more practical solution in this scenario through its simplified structure and lower computational complexity.
>
> **References**
>
> [1] Veličković, Petar, et al. "Graph attention networks." arXiv preprint arXiv:1710.10903 (2017).
>
> [2] Hamilton, Will, Zhitao Ying, and Jure Leskovec. "Inductive representation learning on large graphs." Advances in neural information processing systems 30 (2017).
>
>
>
> We would like to thank the reviewer again for taking the time to review our paper.

---

> > ### Comment · Reviewer_LZHh · 2024-08-12
> >
> > The authors have addressed all my questions, and they explicitly compare with imputation methods in traffic data, and hence I raised my score.

---

> > > ### Author Response · Authors · 2024-08-12
> > > **Reply to Reviewer LZHh**
> > >
> > > Thank you very much for your positive comments and score improvement on our paper. We are pleased to know that we responded adequately to your questions.

---

### Author Rebuttal · Authors · 2024-08-06

We appreciate the reviewers thoughtful and detailed comments, and agree with the majority of the comments and suggestions. In terms of the overall identified weaknesses,the reviewers' concerns can be roughly grouped into:

1. The need for a clearer explanation of the significance and unique challenges of solving such problem specifically for satellite networks.

2. The need for add a Figure for transfer module and some references for certain equations.

3. Including more baselines to strengthen the evaluation, such as CDSA, SAITS, et.al.

We believe that these three concerns are linked, and get to the heart of what we are attempting to show in this paper.

As a crucial component of future 6G systems, satellite networks provide seamless and efficient communication services for global users. In recent years, satellite networks have received increasing attention and are now under construction. Traffic engineering and topology engineering of satellite networks, such as admission control, routing and congestion control, are key to achieve efficient control of satellite networks, which rely on real-time perception of global traffic data. However, the limitations of satellite networks make real-time global traffic data collection extremely costly and impractical, hindering performance improvements. To address this challenge, we propose Satformer, an accurate and robust traffic estimation method for satellite networks. Satformer can accurately recover global traffic data with just 2% of sampling data, demonstrating strong robustness and significantly reducing deployment costs and data collection overhead. This method facilitates the implementation of efficient control mechanisms in real satellite network systems. Additionally, Satformer aids network administrators in enhancing network status perception and optimizing network operation and maintenance. To the best of our knowledge, this is the first work on satellite network traffic estimation.

The unique challenges of traffic estimation for satellite networks can be summarized as follows: (1) Spatial and temporal dynamics: Satellite networks experience highly dynamic traffic patterns due to the constant movement of satellites, varying regional demands, and differing satellite coverage areas. (2) Sparsity and incompleteness: Traffic data in satellite networks is often sparse and incomplete due to the selective and intermittent nature of communication links and Space weather, radiation, and other environmental factors, making it difficulty to the parameter adjustment of deep learning models. (3) Limited computational resources: Satellites have limited computational resources and power, which can restrict the complexity of the traffic estimation models that can be deployed on them.

The transfer module is illustrated in the attached PDF. We have also provided clearer explanations for specific mathematical symbols and equations. Additionally, we have supplemented Satformer with relevant theoretical proofs.

We further evaluate Satformer by comparing it with CDSA and SAITS. Additionally, we assess the performance of Satformer and baseline models on the real-world datasets Foursquare and PeMS-Bay. The results are presented in Tables I, Ⅱ and Ⅲ of the attached PDF.

---

### Decision · Program_Chairs · 2024-09-25

**Decision:**

Accept (poster)

**Comment:**

The paper received 5/7/7/4 ratings. Reviewers acknowledge the clear presentation and thorough experimental evaluations. Reviewer 4's major concern came from the novelty of the proposed method. The AC recognize the application in satellite networks analysis is still valuable to the community, thus recommend acceptance.